# Sampling-guided Heterogeneous Graph Neural Network with Temporal Smoothing for Scalable Longitudinal Data Imputation

## Abstract

In this paper, we propose a novel framework, the Sampling-guided Heterogeneous Graph Neural Network (SHT-GNN), to effectively tackle the challenge of missing data imputation in longitudinal studies. Unlike traditional methods, which often require extensive preprocessing to handle irregular or inconsistent missing data, our approach accommodates arbitrary missing data patterns while maintaining computational efficiency. SHT-GNN models both observations and covariates as distinct node types, connecting observation nodes at successive time points through subject-specific longitudinal subnetworks, while covariate-observation interactions are represented by attributed edges within bipartite graphs. By leveraging subject-wise mini-batch sampling and a multi-layer temporal smoothing mechanism, SHT-GNN efficiently scales to large datasets, while effectively learning node representations and imputing missing data. Extensive experiments on both synthetic and real-world datasets, including the Alzheimer's Disease Neuroimaging Initiative (ADNI) dataset, demonstrate that SHT-GNN significantly outperforms existing imputation methods, even with high missing data rates (e.g., 80%). The empirical results highlight SHT-GNN's robust imputation capabilities and superior performance, particularly in the context of complex, large-scale longitudinal data.

## 1 Introduction

With the advancement of data collection and analysis techniques, longitudinal data has gained increasing importance across various scientific fields, such as biomedical science, economics, and e-commerce (Pekarčík et al., 2022; Sadowski et al., 2021; Mundo et al., 2021). Observation schedules in longitudinal studies vary widely as some follow regular intervals for each subject while others follow irregular patterns. For example, clinical visits scheduled every two months represent a regular observation schedule, whereas follow-ups at 6 months, 1 year, and 2 years post-treatment is an irregular schedule. Additionally, observation schedules may be either consistent or inconsistent across different subjects—some studies may impose uniform intervals for all participants, while others allow for variability due to individual health conditions or other circumstances.

Imputation of missing values poses a major challenge in longitudinal data analysis, both from theoretical and practical standpoints (Daniels & Hogan, 2008; Little & Rubin, 2019). In fact, in complex longitudinal data analysis, such as those in neuroimaging or electronic health record studies, patient follow-up data is collected over time, often resulting in missing values across variables (cross-sectional missingness) and across time (longitudinal missingness). Ideally, as illustrated in Figure 1, each subject would have observations taken at consistent intervals, with an equal number of observations per subject, and each observation would contain fully observed covariates and response variable. However, when missing data arises at specific time points, this misalignment of observations across time can occur. Furthermore, the absence of entire observations at certain time steps can transform regular observation schedules into irregular ones and consistent schedules into inconsistent ones, significantly complicating longitudinal data analysis.

Given the frequent occurrence of missing measurements in studies with extensive data collection, addressing the missingness of both covariates and response variables is crucial for accurate predictions of the target response variable. For instance, in Alzheimer's disease research, the prediction of key

Figure 1: The ideal format of longitudinal data without missing (left); The regularly and consistently observed longitudinal data with missing in covariate variables and response variable (middle). And the irregular and inconsistent observation schedule in longitudinal data due to missing data (right).

biomarkers often relies on incomplete datasets to assess disease progression. Despite methodological advancements, managing missing data in longitudinal studies presents several ongoing challenges. Firstly, how can we effectively model longitudinal data that is irregularly and inconsistently observed? Secondly, how can we design a cohesive forward pass process for data imputation that accommodates varied observation schedules across subjects? Thirdly, how can we accurately predict target response values in the presence of missing covariate data and seamlessly integrate this process into model training? Fourthly, in the context of large-scale longitudinal studies, how can we ensure the scalability of the missing data imputation method?

In this paper, we address the key challenges by introducing the Sampling-guided Heterogeneous Graph Neural Network (SHT-GNN). Unlike traditional imputation methods, our GNN leverages scalable modeling of complex data structures to effectively learn from irregular and inconsistent longitudinal observations. Our SHT-GNN incorporates three key innovations:

1. We handle longitudinal data by sharing trainable parameters across sampled graphs constructed from subject-wise mini-batches. This sampling-guided process ensures scalability for massive longitudinal datasets.

2. Subject-wise longitudinal subnetworks are constructed by connecting adjacent observations with directed edges, while covariate values are transformed into attributed edges, linking observation nodes with covariate nodes. This structure allows SHT-GNN to flexibly model longitudinal data under arbitrary missing data conditions.

3. We introduce a temporal smoothing mechanism across observations for each subject using a multi-layer longitudinal subnetwork. The novel MADGap statistic controls the smoothness within the subnetwork, balancing temporal smoothing with the specificity of observation node representations.

We conducted extensive experiments on both real data and synthetic data. The results demonstrate that SHT-GNN consistently achieves state-of-the-art performance across different temporal characteristics and performs exceptionally well even under high missing rates for both covariates and response variable. Furthermore, we applied the proposed SHT-GNN model to the Alzheimer's Disease Neuroimaging Initiative (ADNI) dataset to predict the critical biomarker $A\beta 42/40$. SHT-GNN's strong performance in predicting $A\beta 42/40$ is confirmed through validation with ground truth data and in downstream analyses. Additionally, we conduct extensive ablation studies, demonstrating the longitudinal network's ability to perform temporal smoothing through multi-layer longitudinal subnetworks and the critical role of MADGap in guaranteeing variance between observations.

## 2 RELATED WORK

### 2.1 STATISTICAL IMPUTATION METHODS

Statistical methods for imputing missing values in longitudinal data are often derived from multiple imputation techniques originally developed for cross-sectional data. For example, the 3D-MICE method extends the MICE (Multivariate Imputation by Chained Equations) framework to account for both cross-sectional and longitudinal dependencies in data (Luo et al., 2018). Some methods like trajectory mean and last observation carried forward (LOCF) leverage the time-series nature for missing data imputation (Lane, 2008). However, these basic models often fail to capture the complex, nonlinear spatio-temporal dependencies inherent in longitudinal data and lack the scalability

to manage both continuous and discrete covariates simultaneously. The multi-directional multivariate time series (MD-MTS) method seeks to address these limitations by integrating temporal and cross-sectional covariates into a unified imputation framework through extensive feature engineering (Xu et al., 2020). A notable limitation of MD-MTS is its reliance on manually crafted features can be labor-intensive and prone to errors. To overcome such challenges, the time-aware dual-cross-visit (TA-DualCV) method leverages both longitudinal dependencies and covariate correlations using Gibbs sampling for imputation (Gao et al., 2022). Nonetheless, when applied to large-scale longitudinal datasets with numerous repeated observations, those methods involving chained equations and Gibbs sampling can become computationally expensive.

## 2.2 MACHINE LEARNING IMPUTATION METHODS

Machine learning methods for imputing missing values in longitudinal data have primarily utilized recurrent neural networks (RNNs) and their advanced variants, such as long short-term memory (LSTM) networks and gated recurrent units (GRUs). BRITS, for example, leverages bidirectional LSTMs to capture both longitudinal and cross-sectional dependencies by utilizing past and future trends (Cao et al., 2018). Similarly, NAOMI employs a recursive divide-and-conquer approach with bidirectional RNNs to handle missing data imputation (Woillez et al., 2019). CATSI enhances this by combining bidirectional LSTMs with a context vector to account for patient-specific temporal dynamics (Kazijevs & Samad, 2023). A significant limitation of all RNN-based methods, however, is their reliance on consistent time step lengths across different subjects, often requiring padding or truncation to achieve this uniformity. When applied to large datasets, where the number of observations can vary greatly between subjects, padding and truncation may lead to excessive and inefficient computations due to the presence of numerous invalid time steps. Beyond RNN-based approaches, the GP-VAE model integrates deep variational autoencoders with Gaussian processes to address missing data in time series. However, the computational complexity of Gaussian processes, which scales at $O(N^3)$, introduces significant computational bottlenecks when processing large-scale longitudinal data (Fortuin et al., 2020).

## 2.3 GRAPH-BASED IMPUTATION METHODS

Numerous advanced graph-based methods have been developed for data imputation tasks (Zhang & Chen, 2019; Li et al., 2021; Zhang et al., 2023). However, these methods face limitations when handling mixed discrete and continuous features and often fail to capture the temporal dependencies inherent in longitudinal data. For example, RAINDROP employs a graph-guided network to handle irregularly observed time series with varying intervals, effectively addressing missing data imputation and outperforming other methods in downstream prediction tasks (Zhang et al., 2021). Similarly, TGNN4I, which integrates GNNs with gated recurrent units, has shown strong performance in imputing missing data for longitudinal graph data (Oskarsson et al., 2023). Despite their strengths, both RAINDROP and TGNN4I rely on predefined node connections within graph datasets, limiting their applicability to longitudinal tabular data where such connections between subjects do not inherently exist. Additionally, GRAPE demonstrated success using a bipartite graph for feature imputation, but its edge size increases linearly with the number of observations, posing scalability challenges (You et al., 2020). IGRM (Zhong et al., 2023), an extension of GRAPE, constructs a friend network among sample nodes to capture similarities and improve imputation. However, IGRM requires continuous updating of a fully connected graph among all samples, making it computationally expensive and lacking scalability for large-scale longitudinal data.

## 3 PROBLEM FORMULATION

In the longitudinal data structure depicted in Figure 1, the observed variables are divided into two categories: covariates and the response variable. Assume there are $n$ subjects, with the $k$-th subject having $n_k$ observations measured. The total number of observations across all subjects is $N = \sum_{k=1}^{n} n_k$. Typically, the set of observed covariates varies across observations in longitudinal studies. For simplicity, we denote the complete set of covariates that can potentially be observed as $X = \{x_1, \ldots, x_p\}$, where $p$ represents the total number of covariates. Any missing values within this set are treated as missing data. The covariate data for all observations can thus be represented as a matrix $\mathbf{D} = (D_{il}) \in \mathbb{R}^{N \times p}$, where $l \in \{1, \ldots, p\}$ and $i = \sum_{k=1}^{j-1} n_k + m$ indexes the $m$-th

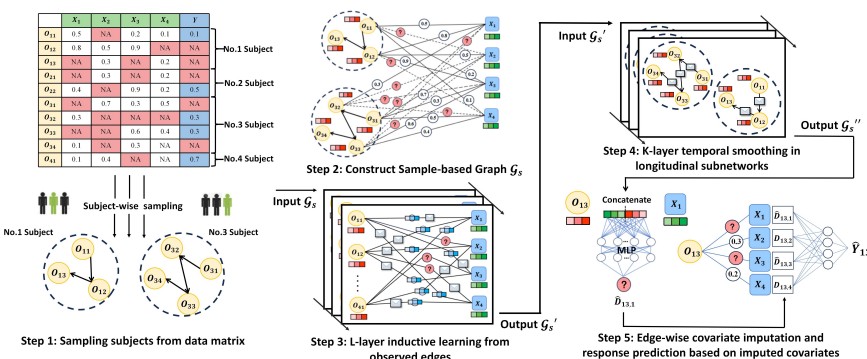

Figure 2: The flow chart for Sampling-guided Heterogeneous Graph Neural Network.

observation of the $j$-th subject. Similarly, the response variable for all observations is denoted as $\mathbf{Y} = (Y_i) \in \mathbb{R}^{N \times 1}$. Given the presence of missing data, we introduce a mask matrix for the covariate data $\mathbf{M}^O = (M_{il}^O) \in \{0, 1\}^{N \times p}$, where $D_{il}$ is observed if $M_{il}^O = 1$. Likewise, the mask matrix for the response variable is denoted by $\mathbf{M}^Y = (M_i^Y) \in \{0, 1\}^{N \times 1}$, where the value of $Y_i$ is observed if $M_i^Y = 1$. Within this irregular or inconsistent longitudinal data structure as depicted in Figure 1, the goal is to predict the response variable $Y_i$ for all $i$ where $M_i^Y = 0$, leveraging the available observation information despite missingness.

## 4 METHOD

We propose SHT-GNN to address the challenges of missing data in irregular or inconsistent longitudinal data mentioned in Section 1. SHT-GNN models the observations and covariates as nodes, connecting them with attributed edges that represent the observed covariate values. Additionally, it employs specially designed longitudinal subnetworks to perform temporal smoothing among observations within the same subject. Initially, observations in longitudinal data can be represented as a graph $\mathcal{G}_L(\mathcal{V}_O, \mathcal{E}_{OO})$ with subject-wise longitudinal subnetworks, where the observation node set $\mathcal{V}_O = \{u_1, u_2, \ldots, u_N\}$ represents all observations. The observation nodes that belong to the same subject form their own independent longitudinal subnetwork. In each longitudinal subnetwork, observation nodes at adjacent time points are connected by directed edges. The directed edge set in $\mathcal{G}_L$ is denoted as $\mathcal{E}_{OO} = \{(u_i, u_{i'}, e_{i \to i'})|u_i, u_{i'} \in \mathcal{V}_O, S(u_i) = S(u_{i'}), u_i \prec u_{i'}\}$, where $S(u)$ denotes the subject that observation $u$ belong to, and $u_i \prec u_{i'}$ represents $u_i$ is the direct predecessor of $u_{i'}$ in terms of time.

In irregularly and inconsistently observed longitudinal observation data, each observation may have measured different covariates. SHT-GNN establishes a covariate node set $\mathcal{V}_C = \{v_1, v_2, \ldots, v_p\}$, and construct edges between observation nodes and covariate nodes to represent the observed covariate values. The longitudinal data matrix $\mathbf{D} \in \mathbb{R}^{N \times p}$ and the missing indicator matrix $\mathbf{M}^O \in \{0, 1\}^{N \times p}$ can be represented as a bipartite graph $\mathcal{G}_B = (\mathcal{V}, \mathcal{E}_{OC})$. The undirected edge set $\mathcal{E}_{OC} = \{(u_i, v_l, e_{u_i v_l})|u_i \in \mathcal{V}_O, v_l \in \mathcal{V}_C, M_{il}^O = 1\}$, where the edge feature $e_{u_i v_l}$ takes the value of the corresponding feature $e_{u_i v_l} = D_{il}$. Then a longitudinal dataset can be represented as the union of one undirected bipartite graph and multiple subject-wise longitudinal subnetworks: $\mathcal{G}(\mathcal{V}, \mathcal{E}) = \mathcal{G}_B \cup \mathcal{G}_L = \mathcal{G}(\mathcal{V}_O \cup \mathcal{V}_C, \mathcal{E}_{OO} \cup \mathcal{E}_{OC})$. After constructing SHT-GNN, the task of response variable prediction under missing covariate imputation can be represented as learning the mapping: $Y_i = [f(\mathcal{G})]_i$ by minimizing the difference between $Y_i$ and $\widehat{Y}_i$, for all $i$ where $M_i^Y = 1$.

### 4.1 LEARNING IN SAMPLING-GUIDED HETEROGENEOUS GRAPH NEURAL NETWORK

#### 4.1.1 SUBJECT-WISE MINI-BATCH SAMPLING

It is evident that the spatial complexity of graph $\mathcal{G}(\mathcal{V}, \mathcal{E})$ increases with the number of subjects. When working with longitudinal data, it is common to encounter a large number of repeated observations or subjects. When dealing with millions or even tens of millions of observations, directly training a huge graph composed of all subjects is impractical and inefficient (You et al., 2020; Zhong et al., 2023).

SHT-GNN performs subject-wise mini-batch sampling at the beginning of each training phase. The corresponding graph is then constructed based on the sampled subjects for the current training phase. In SHT-GNN, the graphs associated with each training phase share the same trainable parameters.

Specifically, assume there are $n$ subjects, represented by the subject set $S = \{1, 2, \ldots, n\}$, where the $k$-th subject corresponds to an observation set $O_k$. Across all observation sets, there are a total of $N$ observations. At the beginning of each training phase, $s$ subjects are randomly sampled from $n$ subjects to form a subject-wise mini-batch $S_0 = \text{Sample}(S, s)$, yielding the corresponding observation-wish mini-batch $O_s = \bigcup_{k \in S_0} O_k$. The observations in $O_s$ are then employed to construct the graph $\mathcal{G}_s = (\mathcal{V}_s, \mathcal{E}_s)$, where $\mathcal{V}_s = \mathcal{V}_{O_s} \cup \mathcal{V}_C$, with $\mathcal{V}_{O_s} = \{u_i | i \in O_s\}$. The edge set $\mathcal{E}_s$ is constructed within $\mathcal{V}_s$ according to the definitions provided above. Assuming the trainable parameters in SHT-GNN are denoted by $\theta$, and the loss function under input graph $\mathcal{G}_{\text{input}}$ is $L(\mathcal{G}_{\text{input}}, \theta)$. The parameter learning process on the sampled graph in each sampling phase can be expressed as:

$$\theta^{(t+1)} = \theta^{(t)} - \eta_t \nabla_\theta L_{\mathcal{S}_0}(\mathcal{G}_s, \theta^{(t)}), \tag{1}$$

where $L_{\mathcal{S}_0}(\mathcal{G}_s, \theta^{(t)})$ represents the loss function obtained from the forward pass in graph $\mathcal{G}_s$, and $\eta_t$ represents the learning rate. Here the forward pass in $\mathcal{G}_s$ begins with inductive learning within a multi-layer bipartite graph to derive representations of observation and covariate nodes. Afterward, temporal smoothing is conducted within the longitudinal subnetwork of each subject.

### 4.1.2 INDUCTIVE LEARNING IN MULTI-LAYER BIPARTITE GRAPH

In SHT-GNN, all the information from the observed data is derived from the attributed edges connecting observation nodes and covariate nodes. The representations of all nodes need to be inductively learned from these edges. Inspired by GraphSAGE, which is a variant of GNNs known for its strong inductive learning capabilities. SHT-GNN modifies GraphSAGE architecture by introducing edge embeddings in the bipartite graph. At the $l$-th layer in $\mathcal{G}_B$, the message generation function takes the concatenation of the embedding of the source node $\mathbf{h}_v^{(l-1)}$ and the edge embedding $\mathbf{e}_{uv}^{(l-1)}$ as input:

$$\mathbf{m}_v^{(l)} = \text{AGG}_l \left( \sigma \left( \mathbf{P}^{(l)} \cdot \text{CONCAT}(\mathbf{h}_u^{(l-1)}, \mathbf{e}_{uv}^{(l-1)}) \right) \mid \forall u \in \mathcal{N}(v, \mathcal{E}_s) \right) \quad \forall v \in \mathcal{V}_s, \tag{2}$$

where $\text{AGG}_l$ is message aggregation function for all node, $\sigma$ is the non-linearity activation function, $\mathbf{P}^{(l)}$ is the trainable weight matrix and $\mathcal{N}$ is the node neighborhood function. Subsequently, all nodes update their embeddings based on the concatenation of aggregated messages and their local representations:

$$\mathbf{h}_v^{(l)} = \sigma \left( \mathbf{Q}^{(l)} \cdot \text{CONCAT}(\mathbf{h}_v^{(l-1)}, \mathbf{m}_v^{(l)}) \right) \quad \forall v \in \mathcal{V}_s, \tag{3}$$

where $\mathbf{Q}^{(l)}$ is the trainable weight matrix. Subsequently, the edge embeddings are then updated based on the updated node embeddings at both ends of each edge:

$$\mathbf{e}_{uv}^{(l)} = \sigma \left( \mathbf{W}^{(l)} \cdot \text{CONCAT}(\mathbf{e}_{uv}^{(l-1)}, \mathbf{h}_u^{(l-1)}, \mathbf{h}_v^{(l-1)}) \right) \quad \forall \mathbf{e}_{uv} \in \mathcal{E}_s, \tag{4}$$

where $\mathbf{W}^{(l)}$ is the trainable weight matrix.

### 4.1.3 TEMPORAL SMOOTHING IN MULTI-LAYER LONGITUDINAL SUBNETWORKS

Temporal smoothing is a crucial technique in the imputation of longitudinal data because it leverages the temporal correlation within the observations. After $L$ layers of inductive learning in the bipartite graph, SHT-GNN innovatively performs temporal smoothing through $K$ layers of message passing and representation updates within the subject-wise longitudinal subnetworks. At the $k$-th layer of each subject-wise longitudinal subnetwork, the message passing function for observations $u_i$ to $u_{i'}$ takes the source node embedding $\mathbf{h}_{u_i}^{(L+k)}$ and the edge weight $w_{u_i \to u_{i'}}^{(L+k)}$ as input:

$$\mathbf{m}_{u_i \to u_{i'}}^{(L+k)} = \mathbf{h}_{u_i}^{(L+k)} \cdot w_{u_i \to u_{i'}}^{(L+k)}, \text{ where } S(u_i) = S(u_{i'}), u_i \prec u_{i'}, \forall u_i, u_{i'} \in \mathcal{V}_{O_s}, \tag{5}$$

in which $S(u)$ denotes the subject that observation $u$ belong to and $u_i \prec u_{i'}$ represents that $u_i$ is the direct predecessor of $u_{i'}$ in terms of time. For each pair of message passing described in (5), the target observation node $u_{i'}$ updates its representation as follows:

$$\mathbf{h}_{u_i}^{(L+k+1)} = \sigma \left( \mathbf{U}^{(L+k)} \cdot \text{CONCAT}(\mathbf{h}_{u_{i'}}^{(L+k)}, \mathbf{m}_{u_i \to u_{i'}}^{(L+k)}) \right), \tag{6}$$

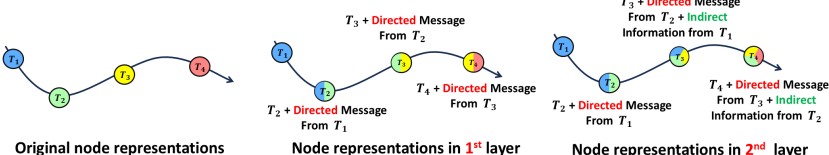

Figure 3: The variation of information in the representation of observation nodes during multi-layer message passing and representation updates for temporal smoothing within longitudinal subnetworks.

where $\mathbf{U}^{L+k}$ is the trainable parameter matrix for embedding updates. In (5), the weight for edges connecting observation nodes needs to be calculated and updated during training. The edge weight is computed as follows:

$$w_{u_i \to u_{i'}}^{(L+k)} = D_{u_i \to u_{i'}} \cdot J_{u_i \to u_{i'}} \cdot \text{Cos}\left(\mathbf{h}_{u_i}^{(L+k)}, \mathbf{h}_{u_{i'}}^{(L+k)}\right). \tag{7}$$

Here, $D_{u_i \to u_{i'}}$ represents the time decay weight in SHT-GNN. When facing irregular observation schedules in longitudinal data, the time interval between observations $u_i$ and $u_{i'}$ is not fixed. We employ exponential decay functions to compute the time decay weights in (7). We define

$$D_{u_i \to u_{i'}} = \exp\left(-\frac{|T(u_i) - T(u_{i'})|}{\Delta_{max}}\right) \quad (8) \qquad \text{and} \qquad J_{u_i \to u_{i'}} = \gamma - \frac{|\mathbb{A} \cap \mathbb{B}|}{|\mathbb{B}|}. \tag{9}$$

In (8), $T(u)$ represents the time step associated with observation $u$, and $\Delta_{max}$ represents the longest time interval between adjacent observations for the current subject. We further introduce the Jaccard distance between the sets of observed covariates in the weight calculation for message passing. Suppose the covariate sets observed by $\mathbf{h}_{u_i}$ and $\mathbf{h}_{u_{i'}}$ are denoted as $\mathbb{A}$ and $\mathbb{B}$, respectively. As shown in (9), a larger $J_{u_i \to u_{i'}}$ implies that observation $u_i$ contains more covariates that are not observed in observation $u_{i'}$, and $\gamma$ is a hyperparameter constant for preventing the weights from becoming zero. Therefore, $\mathbf{h}_{u_{i'}}$ need to borrow more information from $\mathbf{h}_{u_i}$. Beyond the time decay weight and Jaccard distance, the strength of message passing for a specific observation $\mathbf{h}_{u_{i'}}$ from its immediate predecessor $\mathbf{h}_{u_i}$ is intuitively determined by the similarity between the currently observed covariates of $\mathbf{h}_{u_i}$ and $\mathbf{h}_{u_{i'}}$. SHT-GNN then introduces the cosine similarity between $\mathbf{h}_{u_i}$ and $\mathbf{h}_{u_{i'}}$ in the edge weight as discribed in (7).

**Intuitions on why longitudinal subnetworks work**. In the SHT-GNN, multi-layer message passing and representation updates within longitudinal subnetworks enable observations from the same subject to leverage information from previous time steps. As shown in Figure 3, in a two-layer longitudinal subnetwork, assume there are time-ordered observations $u_{t_1}, u_{t_2}, u_{t_3} and u_{t_4}$. The message passing and representation update in the first layer allow $u_{t_m}$ to directly draw information from $u_{t_{m-1}}$. Subsequently, while the temporal smoothing in the second layer enable $u_{t_m}$ to indirectly capture information from $u_{t_{m-2}}$ via the representation of $u_{t_{m-1}}$. This iterative process can be extended with additional layers, allowing observation nodes to integrate information from progressively earlier time steps, thereby complete temporal smoothing in subject-wise longitudinal subnetworks.

### 4.1.4 COVARIATE IMPUTATION IN SHT-GNN

After conducting $L$ layers of inductive learning on the bipartite graph and $K$ layers of temporal smoothing within longitudinal subnetworks, edge-level predictions are made at the $(L + K)$-th layer:

$$\widehat{D}_{il} = \mathbf{O}_{\text{impute}}\left(\text{CONCAT}(\mathbf{h}_{u_i}^{(L+K)}, \mathbf{h}_{v_l}^{(L+K)})\right) \quad \forall u_i \in V_{O_s}, \ v_l \in V_F,$$

where $\mathbf{O}_{\text{impute}}$ is a multilayer perceptron (MLP). Here, $\widehat{D}_{il}$ represents the imputation output for the $l$-th covariate for the $i$-th observation.

### 4.1.5 RESPONSE VARIABLE PREDICTION

Finally, we complete the prediction of the response variable based on the imputed covariates:

$$\widehat{Y}_i = \mathbf{O}_{\text{predict}}\left(\text{CONCAT}(\widehat{D}_{i1}, \widehat{D}_{i2}, ..., \widehat{D}_{ip})\right) \quad \forall u_i \in \mathcal{V}_{O_s},$$

where $\mathbf{O}_{\text{predict}}$ is a multilayer perceptron and $\widehat{Y}_i$ represents the predicted response.

### 4.1.6 Loss function for SHT-GNN

In both the response variable prediction and covariate imputation tasks, the loss function takes the following form:

$$\text{Loss} = \text{MSE} - \lambda \cdot \text{MADGap}.$$

Here, $\text{MSE} = (\sum_{i=1}^{N} M_i^Y)^{-1} \sum_{i=1}^{N} M_i^Y \cdot (\widehat{Y}_i - Y_i)^2$, where $M_i^Y$ is the missing indicator. MADGap (Mean Average Distance Gap) is a statistical measure used to quantify the degree of over-smoothing in GNNs (Chen et al., 2020). A large MADGap value indicates that the node receives more useful information than noise. In SHT-GNN, the multi-layer longitudinal subnetwork between observation nodes represents the process of temporal smoothing. However, multi-layer message passing can lead to over-smoothing, causing the embedding representations of different observations within the same subject to become overly similar. SHT-GNN addresses this by maximizing MADGap to mitigate over-smoothing in the GNN. MADGap is defined for individual nodes as follows:

$$\text{MADGap} = \text{MAD}_{\text{remote}} - \text{MAD}_{\text{neighbour}}.$$

In SHT-GNN, for the $k$-th subject's subnetwork with $n_k$ time ordered observations $\{u_1, u_2, ..., u_{n_k}\}$, MADGap is calculated as:

$$\text{MADGap}_k = \frac{1}{n_k} \sum_{m=1}^{n_k} \mathbf{1}_{m>2} \cdot \left( \frac{1}{m-2} \sum_{m'=1}^{m-2} \text{Cos}(\mathbf{h}_{u_{m'}}, \mathbf{h}_{u_m}) - \text{Cos}(\mathbf{h}_{u_m}, \mathbf{h}_{u_{m-1}}) \right).$$

Here $\frac{1}{m-2} \sum_{m'=1}^{m-2} \text{Cos}(\mathbf{h}_{u_m}, \mathbf{h}_{u_{m'}})$ denotes the similarity between the representations of the $m$-th observation and its past ancestor observation nodes in the longitudinal subnetwork, where $\text{Cos}(\mathbf{h}_{u_m}, \mathbf{h}_{u_{m'}})$ represents the cosine similarity between $\mathbf{h}_{u_m}$ and $\mathbf{h}_{u_{m'}}$. Then for all $n$ subjects:

$$\text{MADGap} = \frac{1}{n} \sum_{k=1}^{n} \left[ \frac{1}{n_k} \sum_{m=1}^{n_k} \mathbf{1}_{m>2} \cdot \left( \frac{1}{m-2} \sum_{m'=1}^{m-2} \text{Cos}(\mathbf{h}_{u_{m'}}, \mathbf{h}_{u_m}) - \text{Cos}(\mathbf{h}_{u_m}, \mathbf{h}_{u_{m-1}}) \right) \right].$$

## 5 Experiment

### 5.1 Baselines

We consider eight baseline methods as follows. 1. Mean (Huque et al., 2018): imputes missing values using the covariate-wise mean. 2. Copy-mean Last Observation Carried Forward (LOCF) (Jahangiri et al., 2023): first imputes using the LOCF, then refines the results based on the population's mean trajectory. 3. Multivariate Imputation by Chained Equations (MICE)(Van Buuren & Groothuis-Oudshoorn, 2011): employs multiple regressions to model each missing value conditioned on other observed covariate values. 4. 3D-MICE (Kazijevs & Samad, 2023): combines MICE and Gaussian processes for longitudinal data imputation. 5. GRAPE: handles feature imputation through graph representation learning. (You et al., 2020) 6. CASTI (Yin et al., 2020): handles missing data in longitudinal data by employing bidirectional LSTM and MLPs. 7. IGRM (Zhong et al., 2023): enhances feature imputation by leveraging the similarity between observations. 8. Transformer (Zeng et al., 2023): handles missing data imputation in time series with self-attention mechanism. 9. GP-VAE (Fortuin et al., 2020): enhances data imputation with Gaussian Process Variational Autoencoder. 10. CTA (Wi et al., 2024): enhances feature imputation via continuous-time Autoencoders.

### 5.2 Experiment on Synthetic Data Simulated from Real Data

**Real data introduction.** To comprehensively evaluate the performance of various methods on longitudinal data with diverse temporal characteristics, we first conduct experiments using synthetic data simulated from real-world datasets. We selected the longitudinal behavior modeling (GLOBEM) dataset as the basis for our synthetic data. The GLOBEM dataset is an extensive, multi-year collection derived from mobile and wearable sensing technologies. It contains data from 497 unique subjects, with over 50,000 observations and more than 2,000 covariates, including phone usage, Bluetooth scans, physical activity, and sleep patterns. For simulating synthetic datasets, all covariate values are sampled directly from the original data. Since the GLOBEM dataset does not include a predefined response variable, we simulate response values based on the observed covariates to construct complete synthetic datasets.

**Response Variable simulation based on real data.** In line with common assumptions in longitudinal data studies, when simulating the response values for the $m$-th observation of the $k$-th subject, assume the response value $Y_i$, where $i = \sum_{k=1}^{j-1} n_k + m$, lies within the $t$-th temporal smoothing window. For simplicity of expression, denote it as $Y_{km}^t$, which is assumed to follow a normal distribution $N(\mu_k^t, \sigma_k^{t\,2})$. Then $\mu_k^t$ and $\sigma_k^{t\,2}$ respectively represent the mean and variance of the response variable for the $t$-th temporal smoothing window for the $k$-th subject. In practice, suppose the $k$-th subject has $n_k$ observations within the time step set $T = \{1, \dots, n_k\}$. We first divide $T$ into $W$ temporal smoothing windows. For the $m$-th observation, which occurs within the $t$-th temporal smoothing window, the response value $Y_{km}^t$ is modeled as follows: $Y_{km}^t = \mu_k^t + \epsilon_{km}$, where $\mu_k^t$ represents the mean response value for the $k$-th subject within the $t$-th temporal smoothing window, and $\epsilon_{km}$ denotes the random fluctuations for the $m$-th observation.

**Response variable simulation execution.** In our experiment, we run multiple random trials under a set of fixed parameters, reporting the average performance of all methods across these trials. In each random trial, we first randomly select $p$ dimensions from the 2000 covariates in the GLOBEM dataset, denoted as $X = \{x_1, x_2, ..., x_p\}$. We then simulate the response values for each observation based on a specified model $f(X) = f(x_1, x_2, ..., x_p)$, which incorporates both linear and nonlinear elements. The details about two different specified models $f(X)$ are provided in Appendix A.1. According to the longitudinal effect model $Y_{km}^t = \mu_k^t + \epsilon_{km}$ described above, we first impute the mean value of the response variable for observations belonging to the same subject and within the same temporal smoothing window. Next, random fluctuations $\epsilon_{km}$ are sampled from the distribution $N(0, \epsilon)$ and incorporated into the formula $Y_{km}^t = \mu_k^t + \epsilon_{km}$ to simulate the final response values.

**Experimental Procedure.** After obtaining complete synthetic datasets, we first split all subjects into a training set and a test set according to ratio $r$. By extracting the observations of all subjects, we obtain the covariate matrix $\mathbf{D}^{\text{train}} \in \mathbb{R}^{N_{\text{train}} \times p}$ and response vector $\mathbf{Y}^{\text{train}} \in \mathbb{R}^{N_{\text{train}}}$ for training, as well as $\mathbf{D}^{\text{test}} \in \mathbb{R}^{N_{\text{test}} \times p}$ and $\mathbf{Y}^{\text{test}} \in \mathbb{R}^{N_{\text{test}}}$ for testing. We respectively generate missing indicator matrices $\mathbf{M}^{\text{train}}$ and $\mathbf{M}^{\text{test}}$ for $\mathbf{D}^{\text{train}}$ and $\mathbf{D}^{\text{test}}$ according to the missing ratio $r_X$, along with the missing indicator vector $\mathbf{V}^{\text{train}}$ for $\mathbf{Y}^{\text{train}}$ based on $r_Y$. For all methods, we employ $\{D_{il}^{\text{train}} | M_{il}^{\text{train}} = 1\}$ and $\{Y_i^{\text{train}} | V_i^{\text{train}} = 1\}$ as input for training. In the testing phase, we use $\{D_{il}^{\text{test}} | M_{il}^{\text{test}} = 1\}$ as input to predict all missing response values $\{\widehat{Y}_i^{\text{test}} | V_i^{\text{test}} = 0\}$. Finally, we evaluate the response variable using root mean square error (RMSE) between $Y_i^{\text{test}}$ and $\widehat{Y}_i^{\text{test}}$ for all $i$ under $V_i^{\text{test}} = 0$.

**SHT-GNN configurations.** We train SHT-GNN for 20 sampling phases with a sampling size of 200. For each sampled graph, we run 1500 training epochs using the Adam optimizer with a learning rate of 0.001(Kingma & Ba, 2024). We employ a three-layer bipartite graph and two-layer longitudinal subnetworks for all subjects. We use the ReLU activation function as the non-linear activation function. The dimensions of both node embeddings and edge embeddings are set to 32. The message aggregation function $\text{AGG}_l$ is implemented as a mean pooling function $\text{MEAN}(\cdot)$. Both $\mathbf{O}_{impute}$ and $\mathbf{O}_{predict}$ are implemented as multi-layer perceptrons (MLP) with 32 hidden units. The $\lambda$ in loss function is set to 0.001.

**Baseline implementation.** For Dicision Tree, GRAPE and IGRM, we directly conduct response prediction under missing covariate matrix. For other baselines, as no end-to-end response prediction approach is available, we first perform covariate imputation using the baselines, followed by utilizing a Multilayer Perceptron (MLP) as the prediction model. To ensure a fair comparison, we apply the same dimensional settings for these MLPs as those used in SHT-GNN.

**Results in medium-dimensional covariates and moderate missing ratios.** Following the experimental procedure described earlier, we set the covariate dimension $p$ to 50. For the window size $w$ and the variance of random fluctuation $\sigma(\epsilon)$, we use the parameter combinations $\{w = 3, \sigma(\epsilon) = 0.1\}$, $\{w = 5, \sigma(\epsilon) = 0.1\}$, and $\{w = 7, \sigma(\epsilon) = 0.2\}$. For each configuration, we first split the subjects with a test ratio of $r = 0.2$, then apply two missing ratio settings to the covariate matrix and response vector: $\{r_X = 0.3, r_Y = 0.3\}$ and $\{r_X = 0.3, r_Y = 0.5\}$. We run All methods for 5 random trials per setting, and report the average RMSE along with its standard deviation for the response prediction on the test set. As shown in Table 1, SHT-GNN outperforms all baselines across all settings, achieving an average reduction of 17.5% in prediction RMSE compared to the best baseline. Across all methods, performance noticeably decline as the temporal smoothing window size and variance increase, particularly for LOCF, Decision Tree, and IGRM. This indicates that the various temporal smoothing characteristics in longitudinal data pose significant challenges for these methods. By

Table 1: Performance comparison with different methods under varying temporal smoothing window sizes and covariate missing ratios. All RMSE values are 0.1 of the actual values.

| Missing ratio | $r_X = 0.3, r_Y = 0.3$ | | | $r_X = 0.3, r_Y = 0.5$ | | |
|---|---|---|---|---|---|---|
| Window size | $w = 3$ | $w = 5$ | $w = 7$ | $w = 3$ | $w = 5$ | $w = 7$ |
| Variance | $\sigma = 0.1$ | $\sigma = 0.15$ | $\sigma = 0.2$ | $\sigma = 0.1$ | $\sigma = 0.15$ | $\sigma = 0.2$ |
| Mean | 0.693±0.009 | 0.826±0.011 | 0.936±0.011 | 0.820±0.012 | 0.833±0.012 | 1.051±0.016 |
| LOCF | 0.786±0.021 | 0.813±0.017 | 0.903±0.028 | 0.772±0.024 | 0.787±0.018 | 0.920±0.021 |
| MICE | 0.724±0.051 | 0.851±0.042 | 0.978±0.038 | 0.825±0.051 | 0.863±0.052 | 0.920±0.041 |
| 3D MICE | 0.689±0.031 | 0.753±0.037 | 0.847±0.029 | 0.741±0.035 | 0.785±0.037 | 0.883±0.021 |
| GRAPE | 0.671±0.013 | 0.786±0.020 | 0.865±0.034 | 0.765±0.013 | 0.799±0.027 | 0.935±0.037 |
| CATSI | 0.701±0.034 | 0.732±0.026 | 0.832±0.023 | 0.749±0.047 | 0.748±0.038 | 0.885±0.027 |
| IGRM | 0.682±0.015 | 0.768±0.011 | 0.874±0.013 | 0.786±0.034 | 0.831±0.020 | 0.928±0.012 |
| GP-VAE | 0.733±0.011 | 0.793±0.018 | 0.851±0.021 | 0.731±0.023 | 0.769±0.025 | 0.933±0.030 |
| Transformer | 0.611±0.022 | 0.691±0.023 | 0.791±0.013 | 0.678±0.013 | 0.718±0.019 | 0.873±0.021 |
| CTA | 0.581±0.010 | 0.678±0.010 | 0.778±0.019 | 0.653±0.015 | 0.673±0.013 | 0.835±0.019 |
| **Our Method** | **0.552±0.011** | **0.650±0.014** | **0.759±0.018** | **0.623±0.013** | **0.653±0.018** | **0.818±0.020** |

integrating inductive learning with temporal smoothing, SHT-GNN demonstrate stable and superior performance in these scenarios.

**Results in high-dimensional covariates and high missing ratios.** We also compare the performance of different methods under higher covariate dimensions and missing ratios. As shown in Table 4 (Appendix A.3), SHT-GNN consistently outperforms all baselines across different settings, achieving an average reduction of 18% in prediction RMSE compared to the best baseline.

## 5.3 ALZHEIMER'S DISEASE NEUROIMAGING INITIATIVE STUDY DATASET

**ADNI dataset introduction.** We apply SHT-GNN to the real data from Alzheimer's Disease Neuroimaging Initiative (ADNI) study. We propose our model to predict the CSF biomarker Amyloid beta 42/40 ($A\beta 42/40$), which has been demonstrated as a crucial biomaker in ADNI study. ADNI dataset containing 1,153 subjects and 10,033 observations. The covariate matrix has 83 dimensions, with a missing ratios of 0.32 for covariate matrix and a missing ratio of 0.83 for $A\beta 42/40$. More details about ADNI dataset can be found in Appendix A.5,

**SHT-GNN configurations for ADNI dataset.** For the ADNI dataset, the configurations of SHT-GNN are detailed in Appendix A.6.

**Validation on observed response value.** We use RMSE to validate the prediction accuracy. As shown in Table 2, our proposed SHT-GNN significantly outperforms other methods in predicting $A\beta 42/40$, with an mean 15.5% improvement in accuracy compared to the best baseline.

**Validation on diagnostic labels.** In the previous section, we evaluated the $A\beta 42/40$ prediction results based on the observed ground truth. However, most observations in ADNI do not have recorded $A\beta 42/40$ values, making it impossible to validate the predictions for these observations using RMSE. In the ADNI study, the ulti-

Table 2: Performance comparison in ADNI dataset. For all methods, we conduct 5-fold cross-validation and report the mean values and standard deviations of the results.

| Method | RMSE | AUC | Accuracy |
|---|---|---|---|
| Mean | 0.112±0.002 | 0.671±0.009 | 0.682±0.008 |
| LOCF | 0.108±0.003 | 0.706±0.008 | 0.717±0.015 |
| MICE | 0.109±0.004 | 0.717±0.021 | 0.701±0.021 |
| 3D MICE | 0.103±0.004 | 0.731±0.017 | 0.721±0.018 |
| GRAPE | 0.106±0.002 | 0.724±0.010 | 0.714±0.011 |
| CATSI | 0.104±0.003 | 0.713±0.017 | 0.708±0.013 |
| IGRM | 0.105±0.002 | 0.721±0.013 | 0.713±0.009 |
| Transformer | 0.110±0.005 | 0.735±0.021 | 0.721±0.020 |
| GP-VAE | 0.112±0.007 | 0.738±0.011 | 0.719±0.017 |
| CTA | 0.101±0.003 | 0.751±0.013 | 0.721±0.016 |
| **Our Method** | **0.090±0.001** | **0.829±0.012** | **0.782±0.011** |

mate goal is to predict disease progression, so we validate the predictions by employing them to classify the diagnostic labels. Each observation is associated with a binary diagnostic label: either AD or no-AD. We combine the predicted $A\beta 42/40$ values with fully observed demographic features and build a logistic model for classification. The classification performance can reflect the performance

of $A\beta42/40$ predictions. As shown in Table 2, using SHT-GNN-predicted $A\beta42/40$ combined with basic individual features yields an AUC of 82.92% for the no-AD vs. AD classification.

## 5.4 EXPERIMENTS ON MORE REAL DATASETS

We compared the performance of our method on more longitudinal data and time series, including the AirQuality, Electricity, Energy and PhysioNet ICU datasets. The results in Appendix A.7 shows SHT-GNN's efficiency in borrowing information across features and observations, which significantly enhances its performance, particularly in the context of irregular longitudinal data imputation. However, our method does not show a significant advantage in one-dimensional time series imputation. Regarding performance under different missing mechanisms, apart from the self-masked MCAR, our method still performs excellently under MAR conditions in ADNI and PhysioNet ICU. Additionally, we provide a theoretical understanding in Appendix A.9 that explains the intuition behind our method and its applicability under MCAR and MAR.

## 5.5 ABLATION STUDY ON THE LAYER COUNT OF LONGITUDINAL SUBNETWORKS

As shown in Figure 3, multi-layer longitudinal subnetworks allow observation nodes to borrow information from previous time steps. To validate that longitudinal subnetworks can borrow longer-term historical information by stacking layers, we applied SHT-GNN with one, two and three layers' longitudinal subnetworks under data simulated under temporal smoothing window sizes $w = 3$, $w = 5$, and $w = 7$.

Table 3: Performance comparison of longitudinal subnetworks with different layer count across different window sizes. All RMSE values are 0.1 of the actual values.

| Window size | $w = 3$ | $w = 5$ | $w = 7$ |
|---|---|---|---|
| None | 0.673±0.019 | 0.757±0.015 | 0.933±0.018 |
| One-Layer | 0.621±0.012 | 0.739±0.015 | 0.889±0.017 |
| Two-Layer | 0.593±0.015 | 0.687±0.019 | 0.827±0.020 |
| Three-Layer | 0.552±0.011 | 0.659±0.014 | 0.763±0.018 |

In all cases, we set the subject test ratio $r$ to 0.2, along with the missing rate of both the covariate matrix and the response vector to 0.3. As shown in Table 3, as $w$ increases, SHT-GNN with three-layer subnetworks outperform the one-layer and two-layer versions by a larger margin. This suggests that additional layers in longitudinal subnetworks enhance performance in data with longer-span temporal smoothing, which demonstrating the multi-layer longitudinal networks designed for temporal smoothing are effectve. Additionally, we conduct an ablation study for the MADGap component in SHT-GNN. The results and more details can be found in Appendix A.4.

## 5.6 SCALABILITY OF SHT-GNN

In longitudinal data, a large number of repeated observations often lead to massive data size, making scalability a critical concern for missing data imputation methods. Specifically, we compare the scalability of SHT-GNN with two cutting-edge GNN-based imputation methods: GRAPE and IGRM. For experimental simplicity, we conduct imputation on datasets with varying observation sizes of $N \times 10$ (where $N = 50, 500, 1000$, and $5000$). In SHT-GNN, we fix the subsample size at 500 subjects, resulting in a total of 5000 observations. For all three methods, we report the memory consumption and forward computation time per epoch during training. The results show that both GRAPE and IGRM exhibit significant increases in memory usage and computation time as observation size grows. However, by fixing the sampled subject batch size, SHT-GNN achieves consistent memory usage and computation time per epoch, regardless of the observation size.

## 6 CONCLUSION

In this paper, we present SHT-GNN, a scalable and accurate framework for longitudinal data imputation. Our method combines a sampling-guided training policy with inductive learning, temporal smoothing, and a custom-designed loss function to address the challenges posed by irregular and inconsistent longitudinal missing data. Compared to state-of-the-art imputation techniques, SHT-GNN consistently improves response prediction across both synthetic and real-world datasets. Extensive experiments confirm the efficacy of our multi-layer longitudinal subnetwork for temporal smoothing.

## 7 REPRODUCIBILITY STATEMENT

In this paper, we used the GLOBEM and ADNI datasets for our experiments. For the GLOBEM dataset, access can be requested at https://physionet.org/content/globem/, which is divided into four groups of subjects. In this study, we use the original data without any preprocessing. For the ADNI dataset, access can be requested at https://ida.loni.usc.edu/. The ADNI study is divided into different phases, and we select those subjects with complete basic genetic observations in the ADNI 1, ADNI 2, and ADNI GO phases. In the supplementary materials, we provide the complete code for the proposed SHT-GNN. The original codes for other baselines can be found through the links to the referenced papers in the Section. In this study, all models are trained on a Windows 10 64-bit OS (version 19045) with 32GB of RAM (AMD Ryzen 7 4800H CPU @ 2.9GHz) and 4 NVIDIA GeForce RTX 2060 GPUs with Max-Q Design.

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

# A APPENDIX

## A.1 RESPONSE VECTOR SIMULATION

In the experiment involving medium-dimensional covariates and moderate missing ratios, we simulate response variables from a 50-dimensional covariate matrix for each observation. Specifically, we use a response simulation model that includes both linear and nonlinear components. The default model for simulating the response variables is as follows:

$$
y = 0.25 \cdot x_1 + 2 \cdot \left( \frac{\log(x_2 + 10)}{25} \right)^2 - 0.4 \cdot x_3 - 0.15 \cdot \left( x_4 + 5 \cdot e^{-5(1.5 - \log(x_5))^2/2} \right)
$$
$$
-0.25 \cdot \log(x_6 + 1) + 0.4 \cdot x_7 + 0.021 \cdot \sin(x_8) + 0.04 \cdot \sqrt{x_9} + 0.1 \cdot e^{x_{10}}
$$
$$
+0.05 \cdot \log(x_{11} + 1) + 0.02 \cdot \tan(x_{12}) + 0.015 \cdot \cos(x_{13})
$$
$$
+0.07 \cdot \log(x_{14} + 1) + 3.5 \cdot \sqrt{x_{15}} + \epsilon
$$

Each $x_i$ is randomly selected without repetition from all the covariates. Before simulating the response values based on the observed covariates, all covariate values are normalized using the MinMax Scaler (Rajaraman, 2011). The term $\epsilon$ represents random noise following a normal distribution $N(0, 0.175)$. In the experiment involving high-dimensional covariates and moderate missing ratios, we simulate response variables from a 100-dimensional covariate matrix for each observation. We use a response simulation model with higher-order inputs and more complex expressions, as shown below:

$$
y = 0.3 \cdot \sqrt{x_1} - 0.4 \cdot x_2^2 + 0.15 \cdot \log(x_3 + 10^{-6}) + 0.2 \cdot \exp(0.5 \cdot x_4) - 0.1 \cdot x_5
$$
$$
+ 0.05 \cdot \sin(2\pi \cdot x_6) + 0.25 \cdot \log 1p(x_7) - 0.1 \cdot \cos(2\pi \cdot x_8) + 0.35 \cdot \tan(\text{clip}(x_9, -0.5, 0.5))
$$
$$
+ 0.05 \cdot \arcsin(\text{clip}(x_{10}, -1, 1)) + 0.2 \cdot x_{11}^3 - 0.3 \cdot \sqrt{x_{12}} + 0.4 \cdot \frac{\log(x_{13} + 1)}{10}
$$
$$
+ 0.15 \cdot \sin(2\pi \cdot x_{14}) - 0.1 \cdot \log 1p(x_{15}) + 0.1 \cdot \exp(x_{16}) - 0.05 \cdot \log(x_{17} + 1)
$$
$$
+ 0.2 \cdot x_{18}^2 + 0.3 \cdot \cos(x_{19}) - 0.07 \cdot \tan(x_{20}) + 0.05 \cdot \frac{\sin(x_{21})}{x_{22} + 1}
$$
$$
+ 0.25 \cdot \log 1p(x_{23}) + 0.15 \cdot \arcsin(\text{clip}(x_{24}, -1, 1)) + 0.1 \cdot x_{25}^3 - 0.05 \cdot \sqrt{x_{26}}
$$
$$
+ 0.07 \cdot \log(x_{27} + 1) + 0.2 \cdot \frac{\tan(x_{28})}{1 + x_{29}^2} - 0.1 \cdot \exp(x_{30})
$$
$$
+ 0.3 \cdot \log(x_{31} + 10) + 0.25 \cdot x_{32} + \epsilon.
$$

Similarly, each $x_i$ is randomly selected without repetition from all the covariates. Before simulating the response values based on the observed covariates, all covariate values are normalized using the MinMax Scaler. The term $\epsilon$ represents random noise following a normal distribution $N(0, 0.2)$.

## A.2 DETAILS FOR BASELINES

**Mean**: For each covariate with missing, we fill in the missing values using the mean of the observed values for that covariate across all observations.

**Copy-mean LOCF**: Following the process described in (Jahangiri et al., 2023), missing values are initially imputed using the LOCF (Last Observation Carried Forward) method within each subject to provide an approximation. Next, the population's mean trajectory is used to further refine these imputed values.

**MICE**: As outlined in (Van Buuren & Groothuis-Oudshoorn, 2011), MICE (Multiple Imputation by Chained Equations) performs multiple imputations by modeling each missing value conditioned on the non-missing values in the data. A maximum of 20 iterations is used during the imputation process.

**3D-MICE**: Following the procedure described in (Luo et al., 2018), MICE is configured to perform cross-sectional imputation with a maximum of 20 iterations. And Gaussian Process Regression is applied longitudinally to the time-indexed data for each feature. The Gaussian Process uses an RBF

kernel combined with a constant kernel (Kernel = $\text{C}(1.0) \times \text{RBF}(1.0)$), and the predictions from the Gaussian Process are averaged with the MICE-imputed values to capture both temporal and cross-sectional patterns.

**GRAPE**: Following the setup in (You et al., 2020), GRAPE is trained for 20,000 epochs using the Adam optimizer with a learning rate of 0.001. We employ two GNN layers with 16 hidden units and ReLU activation. The $\text{AGG}_l$ function is implemented as a mean pooling function MEAN($\cdot$). Both the edge imputation and response prediction neural networks are implemented as linear layers.

**CATSI**: Following the setup in (Yin et al., 2020), CATSI is trained for 3,000 epochs using the Adam optimizer with a learning rate of 0.001. We employ the default settings for MLPs and LSTM models in CASTI to impute covariate values.

**IGRM**: Following the default settings described in (Zhong et al., 2023), IGRM employs three GraphSAGE layers with 64 hidden units for bipartite Graph Representation Learning (GRL), and one GraphSAGE layer for friend network GRL. The Adam optimizer with a learning rate of 0.001 and ReLU activation function is used. For initializing the friend network, observation nodes are randomly connected with $|U|$ edges to form the initial network structure, which is updated every 100 epochs during bipartite graph training.

## A.3 RESULTS IN HIGH-DIMENSIONAL COVARIATES AND HIGH MISSING RATIOS

To further compare the performance of different methods under higher covariate dimensions and higher missing ratios, we set the covariate dimension $p$ to 100. The same settings for temporal smoothing window size $w$ and the variance of random fluctuation $\sigma(\epsilon)$ were applied as before. Additionally, two higher missing ratio settings were employed: $\{r_X = 0.5, r_Y = 0.5\}$ and $\{r_X = 0.5, r_Y = 0.7\}$. All methods were run for 5 random trials per setting, and the average RMSE of response prediction on the test set are recorded. As shown in Table 4, SHT-GNN consistently outperforms all baselines across all settings, achieving an average reduction of 18% in prediction RMSE compared to the best baseline.

Table 4: Performance comparison with different methods under varying temporal smoothing windows and covariate missing ratios. All RMSE values are 0.1 of the actual values.

| Missing ratio | $r_X = 0.5, r_Y = 0.5$ | | | $r_X = 0.5, r_Y = 0.7$ | | |
|---|---|---|---|---|---|---|
| Window size | $w = 3$ | $w = 5$ | $w = 7$ | $w = 3$ | $w = 5$ | $w = 7$ |
| Variance | $\sigma = 0.1$ | $\sigma = 0.15$ | $\sigma = 0.2$ | $\sigma = 0.1$ | $\sigma = 0.15$ | $\sigma = 0.2$ |
| Mean | 0.803±0.015 | 0.907±0.022 | 1.054±0.023 | 0.908±0.019 | 0.893±0.020 | 0.975±0.027 |
| LOCF | 0.751±0.029 | 0.813±0.035 | 0.938±0.035 | 0.825±0.023 | 0.823±0.019 | 0.937±0.019 |
| MICE | 0.797±0.043 | 0.848±0.041 | 1.014±0.059 | 0.837±0.036 | 0.893±0.041 | 0.974±0.065 |
| 3D MICE | 0.745±0.021 | 0.773±0.047 | 0.948±0.045 | 0.795±0.036 | 0.793±0.046 | 0.902±0.046 |
| DT | 0.763±0.017 | 0.798±0.018 | 0.981±0.020 | 0.781±0.019 | 0.794±0.027 | 0.891±0.031 |
| GRAPE | 0.724±0.026 | 0.793±0.015 | 0.952±0.021 | 0.809±0.021 | 0.745±0.021 | 0.944±0.021 |
| CATSI | 0.831±0.029 | 0.725±0.021 | 0.945±0.041 | 0.791±0.031 | 0.755±0.041 | 0.903±0.031 |
| IGRM | 0.795±0.010 | 0.831±0.013 | 0.904±0.014 | 0.785±0.012 | 0.815±0.019 | 0.895±0.021 |
| **Our Method** | **0.632±0.010** | **0.672±0.017** | **0.821±0.014** | **0.658±0.021** | **0.641±0.013** | **0.819±0.020** |

## A.4 ABLATION STUDY FOR MADGAP IN SHT-GNN

In SHT-GNN, the multi-layer longitudinal subnetworks are designed for temporal smoothing. However, the degree of smoothing among observations for the same subject may vary across different longitudinal studies. The loss function incorporates MADGap to promote greater variance amonng observation representations for a given subject, enabling the model to effectively trade off between temporal smoothing and representation diversity. To evaluate the impact of MADGap, we test SHT-GNN with and without MADGap under different temporal smoothing window sizes. As shown in Table 5, the results shows that incorporating MADGap enhances SHT-GNN's performance by an average of 6%. Moreover, as the degree of temporal smoothing increases, the design incorporating MADGap exhibited a more substantial advantage over the one without it. This highlights MADGap's capacity to help SHT-GNN capture the unique characteristics of each observation during imputation.

Table 5: Performance comparison of SHT-GNN with and without MADGap across different temporal smoothing windows and missing ratios. All RMSE values are 0.1 of the actual values.

| | SHT-GNN **without** MADGap | | SHT-GNN **with** MADGap | | |
| | $r_X = 0.3$ $r_Y = 0.3$ | $r_X = 0.5$ $r_Y = 0.5$ | $r_X = 0.3$ $r_Y = 0.3$ | $r_X = 0.5$ $r_Y = 0.5$ | **Enhancement** |
|---|---|---|---|---|---|
| Window size 3 | 0.591±0.011 | 0.691±0.013 | 0.552±0.011 | 0.632±0.010 | 9.68% |
| Window size 5 | 0.687±0.012 | 0.718±0.011 | 0.651±0.014 | 0.672±0.017 | 5.57% |
| Window size 7 | 0.801±0.011 | 0.865±0.011 | 0.769±0.018 | 0.821±0.014 | 4.91% |

## A.5 ADNI DATASET INTRODUCTION

We apply SHT-GNN to the real data from Alzheimer's Disease Neuroimaging Initiative (ADNI) study. ADNI is a multi-centre longitudinal neuroimaging study with the aim of developing effective treatments that can slow or halt the progression of Alzheimer's Disease (AD). The ADNI participants were followed prospectively, with follow-up time points at 3 months, 6 months, then every 6 months until up to 156 months. The ADNI study includes a wide range of clinical data such as cognitive assessments, magnetic resonance imaging (MRI) and cerebrospinal fluid (CSF) biomarkers. Numerous studies show that CSF biomarkers are strong indicators of AD progression, but collecting CSF requires invasive procedures like lumbar puncture, leading to high missing data rates. We propose the SHT-GNN model to predict the CSF biomarker Amyloid beta 42/40 ($A\beta 42/40$), which has been a key biomaker. ADNI dataset containing 1,153 subjects and 10,033 observations. The covariate matrix has 83 dimensions, with a missing ratios of 0.32 for covariate matrix and a missing ratio of 0.83 for $A\beta 42/40$.

## A.6 SHT-GNN CONFIGURATIONS FOR ADNI DATASET

We train SHT-GNN for 10 sampling phases with a sampling size of 200. For each sampled graph, we run 1500 training epochs using the Adam optimizer with a learning rate of 0.001. We employ a three-layer bipartite graph and two-layer longitudinal subnetworks for all subjects. We use the ReLU activation function as the non-linear activation function. The dimensions of both node embeddings and edge embeddings are set to 32. The message aggregation function $\text{AGG}_l$ is implemented as a mean pooling function $\text{MEAN}(\cdot)$. Both $\mathbf{O}_{impute}$ and $\mathbf{O}_{predict}$ are implemented as multi-layer perceptrons (MLP) with 32 hidden units. The $\lambda$ in loss function is set to 0.001.

## A.7 EXPERIMENTS ON MORE REAL DATASETS

Table 6: Comparison of methods on different datasets

| | **AirQuality** | **Electricity** | **Energy** | **PhysioNet-2012** |
|---|---|---|---|---|
| Transformer | 0.220±0.019 | 0.889±0.071 | 0.313±0.018 | 0.190±0.019 |
| GP-VAE | 0.287±0.010 | 0.963±0.056 | 0.401±0.025 | 0.398±0.020 |
| CTA | **0.196±0.012** | **0.767±0.042** | 0.205±0.019 | 0.192±0.016 |
| SAITS | 0.201±0.009 | 0.894±0.051 | 0.301±0.012 | 0.190±0.014 |
| MICE | 0.310±0.023 | 1.319±0.051 | 0.371±0.010 | 0.223±0.021 |
| 3D MICE | 0.293±0.009 | 1.083±0.051 | 0.341±0.011 | 0.209±0.015 |
| GRAPE | 0.267±0.013 | 0.891±0.029 | 0.251±0.009 | 0.203±0.005 |
| CATSI | 0.236±0.019 | 0.849±0.071 | 0.201±0.023 | 0.206±0.013 |
| IGRM | 0.242±0.010 | 0.867±0.045 | 0.231±0.013 | 0.193±0.014 |
| **Our method** | 0.212±0.010 | 0.834±0.025 | **0.183±0.011** | **0.187±0.009** |

On the AirQuality and Electricity datasets, the SHT-GNN method demonstrates inferior performance compared to RNN and VAE-based approaches. This indicates that the SHT-GNN model is not well-suited for long-term, single-object time series, which fall outside its intended application scenario.

However, SHT-GNN achieves state-of-the-art performance on the Energy dataset. Unlike the AirQuality and Electricity datasets, the Energy dataset involves time series with multiple subjects and multidimensional features. SHT-GNN leverages its ability to effectively borrow information across features, which significantly enhances its performance, particularly in the context of irregular longitudinal data imputation.

It is worth emphasizing that it can be observed that our method demonstrates a significant advantage on the PhysioNet dataset, which is also clinical longitudinal follow-up data like the ADNI dataset.

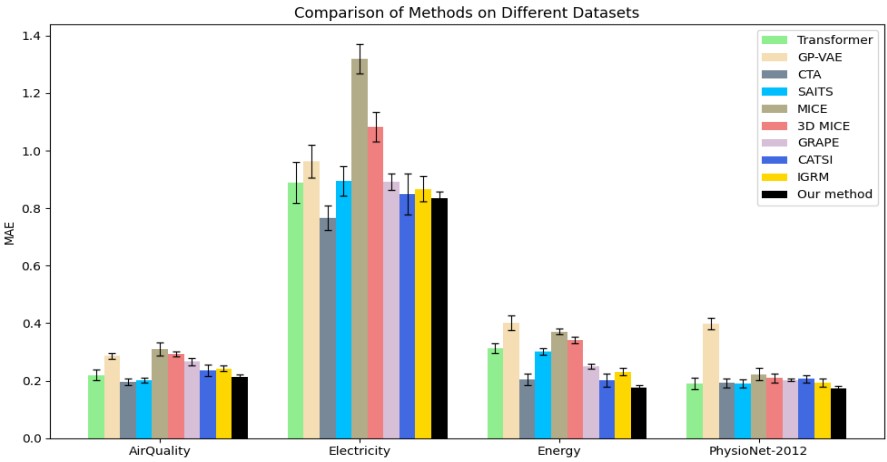

Figure 4: A comparison of the performance of missing data imputation across all methods on additional datasets.

## A.8 SCALABILITY OF SHT-GNN

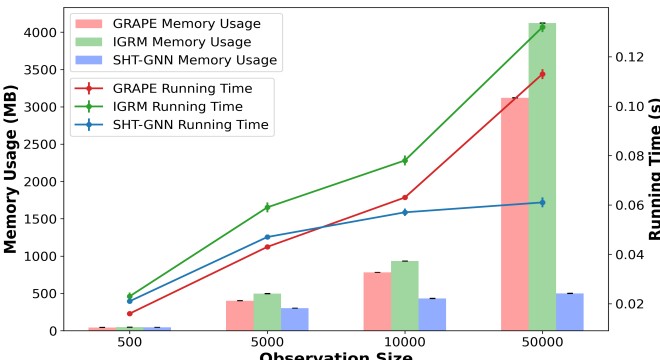

Figure 5: The comparison of scalability under different observation sizes across GNN-based methods.

## A.9 THEORETICAL UNDERSTANDINGS AND INSIGHTS OF SHT-GNN

From an optimization perspective, the missing data imputation process in our method is conceptually similar to Variational Autoencoders (VAEs), where the goal of the reconstruction step is to minimize the reconstruction error as part of the Evidence Lower Bound (ELBO). As is widely recognized, the training objective of a standard VAE for missing data imputation is expressed as (Collier et al., 2020):

$$X_{obs} \xrightarrow[\text{MLPs}]{\text{Encode}} Z \xrightarrow[\text{MLPs}]{\text{Decode}} \hat{X}_{obs}$$

$$\text{Maximize:} \quad \log p(x_{obs}) = \int \log q(z|x_{obs}) \cdot \frac{p(z, x_{obs})}{q(z|x_{obs})} dz \geq \int q(z|x_{obs}) \log \frac{p(x_{obs}, z)}{q(z|x_{obs})} dz$$

$$\text{That is to maximize:} \quad E_{q(z|x_{obs})} \log p(x_{obs}|z) - D_{KL}[q(z|x_{obs}) \parallel p(z)]$$

where $E_{q(z|x)} \log p(x|z)$ represents the reconstruction loss, and $D_{KL}[q(z|x) \parallel p(z)]$ is the regularization term. When maximizing $\log p(x_{obs})$, it is guaranteed that the estimated results for missing data will be consistent in both MCAR and MAR scenarios, a point that has been emphasized in many studies (Mattei & Frellsen, 2019; Collier et al., 2020) .

In our proposed SHT-GNN, we are also theoretically optimizing $\log p(x_{obs})$. Specifically, the calculation and training process can be described as follows:

$$X_{\text{obs}}, Z_O^{\text{init}}, Z_F^{\text{init}} \xrightarrow[\mathcal{G}]{\text{Message Passing, Embedding Update}} Z_O^L, Z_F^L \xrightarrow[\textbf{MLPs}]{\text{Edge-wise Prediction as Missing Data Imputation}} \hat{X}_{\text{obs}}$$

where $X_{\text{obs}}$ denotes the observed values, $Z_O^{\text{init}}$ and $Z_F^{\text{init}}$ represent the initial embedding matrices of all observation and feature nodes, respectively. $Z_O^{\text{L}}$ and $Z_F^{\text{L}}$ denote the embedding matrices of all observation and feature nodes after $L$ layers of forward computation in SHT-GNN. Subsequently, the training objective in SHT-GNN is expressed as:

$$\text{Maximize:} \ \log p(X_{obs}) = \int \log q(Z_O^L, Z_F^L|X_{obs}) \cdot \frac{p(Z_O^L, Z_F^L, X_{obs})}{q(Z_O^L, Z_F^L|X_{obs})} dZ \geq \int q(Z_O^L, Z_F^L|X_{obs}) \log \frac{p(X_{obs}, Z_O^L, Z_F^L)}{q(Z_O^L, Z_F^L|X_{obs})} dZ$$

$$\text{That is to maximize :} \ E_{q(Z_O^L, Z_F^L|X_{obs})} \log p(X_{obs}|Z_O^L, Z_F^L) - D_{\text{KL}}[p(Z_O^L, Z_F^L|X_{obs}) \parallel p(Z_O^L, Z_F^L)]$$

where $E_{q(Z_O, Z_F|X_{obs})} \log p(X_{obs}|Z_O^L, Z_F^L)$ represents the reconstruction loss.

Here, $E_{q(Z_O, Z_F|X_{obs})} \log p(X_{obs}|Z_O^L, Z_F^L)$ is the joint distribution over all observations, which differs from the $E_{q(z|x_{obs})} \log p(x_{obs}|z)$ in VAE. Previously, in the case of VAE, their expectation is calculated on the sample level, typically using the MSE over the observed values of all samples to approximate the reconstruction loss. The specific form is:

$\text{Loss} = \frac{1}{N} \sum_{i=1}^{N} \sum_{j=1}^{p} m_{ij} \cdot (\hat{X}_{ij} - X_{ij})^2$

where $m_{ij}$ is the missing indicator for the $j$-th feature in the $i$-th observation.

In contrast, in SHT-GNN, our reconstruction loss is in the form of a joint distribution, and it is not possible to estimate it by averaging over the samples. This is why we use edge dropout trick (You et al., 2020) in missing data imputation, because directly using the loss over all observed edges will not provide an effective estimate of $E_{q(Z_O, Z_F|X_{obs})} \log p(X_{obs}|Z_O^L, Z_F^L)$. Specifically, we estimate the overall reconstruction loss by randomly calculating the loss on some edges in each batch of different heterogeneous graphs.

In principle, the SHT-GNN and VAE-based methods share conceptual similarities. Furthermore, the reconstruction loss used in both methods, along with the maximized target $p(X_{obs})$, theoretically indicates that our method is also capable of handling missing data in the MAR scenario, in the same way as VAE-based methods.

