# OpenReview forum: "Sampling-guided Heterogeneous Graph Neural Network with Temporal Smoothing for Scalable Longitudinal Data Imputation"
_ICLR.cc/2025/Conference — Submitted to ICLR 2025_

### Official Review · Reviewer_mF3B · 2024-10-30

**Soundness:** 2
**Presentation:** 3
**Contribution:** 2
**Rating:** 5
**Confidence:** 4

**Summary:**

This manuscript proposes a new framework, SHT-GNN, for longitudinal data imputation based on GNN networks.
It uses the bipartite graph to model the relationship between the covariates and observations and uses a direct graph to model the sequential relations of observations.
Modeling the imputation as link prediction in the edge utilizes the graph to aggregate the information for an accurate imputation.
Experiments on both synthetic data and one realistic dataset validate its high performance.

**Strengths:**

S1: The introduction of the method is thoroughly detailed, with complete and comprehensive mathematical formulas provided for each step.

S2: The motivation behind each step is clearly explained and sufficient.

**Weaknesses:**

W1: The novelty of the proposed method is limited.

W1.1. The method utilizes a bipartite graph to model the relationship between observations and covariates, addressing the task of imputation through link prediction. While similar approaches have been widely used for tabular imputation, as seen in models like GRAPE and IGRM, it appears that the author has merely adapted these methods for longitudinal data, which is essentially not very different from standard tabular data. GRAPE and IGRM also utilize the bipartite graph with U to denote the instance and V to denote the feature. The method proposed in this paper also follows this pipeline besides a directed graph used for instances to model the temporal relationship.

W1.2. It utilizes sampling methods for training the graph neural network, a widely used technique for GNNs. However, the sampling strategy employed here does not appear to be specifically designed for longitudinal data.

W1.3. A temporal smoothing module is proposed to connect different observations; however, its effectiveness has not been studied through ablation analysis. Although some experimental results are provided in section 5.4, these experimental results are conducted on synthetic datasets, which already imply strong temporal relationships. In real-world datasets, this relationship may not exist.

=======

W2: The presentations of the work can be enhanced.

W2.1. The figures (Figures 1, 2, 3) are not vector images, causing them to become blurry when zoomed in.

W2.2. In Equation 9, the Jaccard distance (or similarity) is not correctly defined according to the standard formula, so it is inappropriate to call it the “Jaccard distance.” In the Jaccard definition, the denominator should be the union of sets A and B. Referring to it as a “Modified Jaccard Distance” may be more appropriate. You can either correct the formula or explicitly state and justify their modified version of the Jaccard distance.

W2.3: In Equation 6, the left side seems to be $h_{u_{i\prime}}$ rather than $h_{u_{i}}$.

W2.3: There are some typos, Row 223 typo, “observation-wish” -> “observation-wise”; Row 303 typo, “and” inside the formula; Row 416 typo, “dicision tree” -> “decision tree”.

======

W3: Some experiments can be enhanced to better highlight its performance.

W3.1: The simulation process in the generation of the datasets is complex and lacks motivation. As provided in the appendix, a 50-dimensional covariate matrix is constructed for each observation. They use a fixed equation with a lot of fixed numbers (hyperparameters). How these numbers or hyper-parameters are selected and while the usage of this fixed equation needs further explanation or citation.

W3.2: Some default settings of baselines are changed. For example, in the hidden dimension of the GRAPE, it is 64 by default in its original setting but the authors set it to 16 in the experiments (Line 701)

W3.3: The authors adopt two sets of parameters for different datasets, which may affect the model's adaptability and performance across varying data distributions.

W3.4: The authors include some imputation methods for tabular data (e.g., GRAPE and IGRM) and time series (e.g., CASTI) but they are not SOTA. Some more recent imputation methods such as ReMasker for tabular data [1] and [2] for time series are proposed. Coud you please why these specific baselines were chosen, or to include comparisons with the more recent methods like in [1] and [2].

[1]: ReMasker: Imputing Tabular Data with Masked Autoencoding

[2]: Mining of Switching Sparse Networks for Missing Value Imputation in Multivariate Time Series

**Questions:**

Q1: How to apply these tabular data imputation like GRAPE and IGRM for longitudinal data.

Q2: How to apply the time-series data imputation method for multiple observations in longitudinal data.

Q3: How do the state-of-the-art tabular imputation methods and time-series imputation methods perform on the longitudinal data?

---

> ### Author Response · Authors · 2024-11-19
> **Weakness 1: Methodological Innovation and Theoretical Intuition (Part 1)**
>
> Thank you very much for your valuable feedback!
> Indeed, our method shares similarities with existing data imputation approaches based on graph neural networks (GNNs). Specifically, several prior works have utilized bipartite graphs in GNNs for data imputation, including:
>
> [1] You, J., Ma, X., Ding, Y., et al. Handling missing data with graph representation learning. *Advances in Neural Information Processing Systems*, 2020, 33: 19075–19087.
>
> [2] Zhong, J., Gui, N., Ye, W. Data imputation with iterative graph reconstruction. *Proceedings of the AAAI Conference on Artificial Intelligence*, 2023, 37(9): 11399–11407.
>
> [3] Um, D., Park, J., Park, S., et al. Confidence-based feature imputation for graphs with partially known features. *arXiv preprint arXiv:2305.16618*, 2023.
>
> [4] Gupta, S., Manchanda, S., Ranu, S., et al. GRAFENNE: learning on graphs with heterogeneous and dynamic feature sets. *International Conference on Machine Learning*, PMLR, 2023: 12165–12181.
>
> In practice, the aforementioned methods adopt bipartite graph-based imputation frameworks, incorporating advanced modules such as confidence scoring, diffusion models, and transformer mechanisms to capture interdependencies among features. Our approach shares commonalities with these GNN-based methods, particularly in its use of edge-wise prediction for missing data imputation and its reliance on bipartite graph structures.
>
> However, a key distinction lies in our method’s focus on longitudinal data, which is often overlooked in existing approaches. Longitudinal datasets are particularly valuable due to their temporal follow-up structure, yet they frequently encounter significant rates of missing data because of the inherent challenges of long-term studies. None of the aforementioned methods specifically address this unique context.
>
> Our method introduces a temporal smoothing mechanism and a custom-designed loss function tailored to the characteristics of longitudinal data. These innovations enable effective modeling of temporal dependencies and inter-subject variability, setting our approach apart from existing GNN-based imputation methods.
>
> Furthermore, our approach has demonstrated substantial advancements when applied to real-world datasets, such as those for Alzheimer's disease research, highlighting its practical utility. Below, we provide both theoretical and intuitive explanations to demonstrate the novelty of our modules and the necessity of these mechanisms for learning in longitudinal data contexts.

---

> ### Author Response · Authors · 2024-11-19
> **Weakness 1: Methodological Innovation and Theoretical Intuition (Part 2)**
>
> In practice, the process within SHT-GNN can be understood as a form of data reconstruction. From an optimization perspective, this process is conceptually similar to Variational Autoencoders (VAEs), where the goal of the reconstruction step is to minimize the reconstruction error as part of the Evidence Lower Bound (ELBO). As is widely recognized, the training objective of a standard VAE is expressed as:
>
> $$
> X \xrightarrow[\text{MLPs}]{\text{Encode}} Z \xrightarrow[\text{MLPs}]{\text{Decode}} \hat{X}
> $$
>
> $$
> \text{Maximize:} \ \ \log{p(x)} = \int\log{ q(z|x)\cdot{\frac{p{(z,x)}}{q{(z|x)}}}dz} \geq \int q(z|x) \log{\frac{p(x,z)}{q(z|x)}}dz
> $$
>
> $$
> \text{That is to maximize:} \ \ E_{q(z|x)}\log{p(x|z)} - D_{\text{KL}}[q(z|x)\parallel p(z)]
> $$
>
> where $E_{q(z|x)}\log{p(x|z)}$ represents the $\textbf{reconstruction loss}$, and $D_{\text{KL}}[q(z|x)\parallel p(z)]$ is the $\textbf{regularization term}$. In our proposed SHT-GNN, the calculation and training process can be described as follows:
>
> $$
> X_{\text{obs}}, Z^{\text{init}}_{O}, Z_F^{\text{init}}  \xrightarrow[\huge \mathcal{G}]{\text{Message Passing, Embedding Update}} Z^L_O, Z^L_F \xrightarrow[\textbf{MLPs}]{\text{Edge-wise Prediction as Missing Data Imputation}} \hat{X}
> $$
>
> where $X_{\text{obs}}$ denotes the observed values, $Z_{O}^{\text{init}}$ and $Z_{F}^{\text{init}}$ represent the initial embedding matrices of all observation and feature nodes, respectively. $Z_{O}^{\text{L}}$ and $Z_{F}^{\text{L}}$ denote the embedding matrices of all observation and feature nodes after $L$ layers of forward computation in SHT-GNN. Subsequently, the training objective in SHT-GNN is expressed as:
>
> $$
> \text{Maximize:} \ \ \log{p(X_{obs})} = \int\log{ q(Z^L_{O},Z^L_{F}|X_{obs})\cdot{\frac{p{(Z^L_{O},Z^L_{F},X_{obs})}}{q{(Z^L_O,Z^L_F|X_{obs})}}}dZ} \geq \int q(Z^L_O, Z^L_F|X_{obs}) \log{\frac{p(X_{obs},Z^L_O,Z^L_F)}{q(Z^L_O,Z^L_F|X_{obs})}} dZ
> $$
> $\text{That is to maximize}$: $E_{q(Z^L_{O},Z^L_F|X_{obs})}\log{p(X_{obs}|Z^L_{O},Z^L_{F})} - D_{\text{KL}}[p(Z^L_O,Z^L_F|X_{obs}) \parallel p(Z^L_O,Z^L_F)]$
>
> where $E_{q(Z_{O},Z_F|X_{obs})}\log{p(X_{obs}|Z_{O}^L,Z_{F}^L)}$ represents the $\textbf{reconstruction loss}$.
>
> In principle, the SHT-GNN and VAE-based methods share conceptual similarities. However, the key distinctions between the SHT-GNN and VAE-based approaches in the handling of longitudinal data are as follows.
>
> $\textbf{1. Enhanced Representation Capability through Graph Neural Networks (GNNs)}$. In SHT-GNN, the encoding and decoding processes utilize graph neural networks (GNNs) instead of the multilayer perceptrons (MLPs) employed in standard VAEs. GNNs offer significantly stronger representation capabilities by leveraging the inherent data structure and facilitating temporal smoothing across non-independent observations, which is critical in longitudinal settings.
>
> $\textbf{2. Distinct Regularization Term for Non-Independent Observations}$. In VAE-based methods, the regularization term for the latent space distribution, $D_{\text{KL}}[q(z|x) \parallel p(z)]$, quantifies the independent distributions of individual observations. By contrast, SHT-GNN incorporates a KL divergence term, $D_{\text{KL}}[p(Z^L_O,Z^L_F|X_{obs}) \parallel p(Z^L_O,Z^L_F)]$, which measures the joint distribution of all observation representations. To account for dependencies within longitudinal data, SHT-GNN employs multi-layer temporal smoothing and introduces MADGap, enabling improved representation of the joint latent space distribution.
>
> In summary, the foundational intuition behind SHT-GNN lies in optimizing the reconstruction error during the data reconstruction process. Moreover, in the context of longitudinal data, SHT-GNN introduces temporal smoothing and MADGap to strike a balance between temporal consistency and the diversity of observation representations within the latent space.
>
> We have realized that the lack of straightforward presentation and theoretical intuitiveness in our paper has caused difficulties for readers. Therefore, in the revision of our paper, we will include all the aforementioned theoretical understandings as a remark within the article.

---

> ### Author Response · Authors · 2024-11-19
> **Weakness 2,3 : Problems in presentations and Appendix Simulations' reliance to existing studies**
>
> **1. Problems in presentations**
>
> Thank you very much for your valuable suggestions, which are highly important for improving the quality of our paper.
>
> 1.1 Regarding the issue with the figures, we will replace them with vector graphics during the revision process, as long as they do not exceed the maximum upload capacity allowed for the PhD submission system.
>
> 1.2. For the Jaccard distance, you are correct that we made certain modifications to the original Jaccard distance. In its original form, the denominator should be the union of sets A and B, ensuring the distance is symmetric. However, since our message-passing mechanism along the timeline is directional, we aimed to measure the additional information from past observations relative to the current observation. Therefore, we adjusted the denominator to reflect the observation set corresponding to the receiving observation. We acknowledge that this deviates from the original definition, and we greatly appreciate your suggestion. We will adopt your recommendation and revise the term to "adjusted Jaccard distance."
>
> 1.3. Regarding the typos, we sincerely apologize for the errors and the inconvenience they may have caused during your reading. We will correct all the typos you identified and thoroughly review the entire manuscript for spelling and potential representation issues. Once again, thank you for your detailed review and constructive feedback on our paper.
>
> **2. Appendix Simulations and their reliance to existing studies**
>
> In the process of simulation data generation, the response values were generated using a mix of higher-order and lower-order terms, with the parameters and hyperparameters cited from the following references:
>
> [1] Binder, H., Sauerbrei, W., Royston, P. Comparison between splines and fractional polynomials for multivariable model building with continuous covariates: a simulation study with continuous response. *Statistics in Medicine*, 2013, 32(13): 2262–2277.
>
> [2] Wu, H., Zhang, J. T. *Nonparametric regression methods for longitudinal data analysis: mixed-effects modeling approaches*. John Wiley \& Sons, 2006.
>
> Specifically, the generation of continuous response variables followed the approach outlined in [1]. The simulation formula proposed in [1] was designed to ensure that the generated data align with a predefined Structural Causal Model. This was achieved by employing a linear or nonlinear combination of covariates’ lower-order terms (e.g., square root or original values), higher-order terms (e.g., squared or cubed values), and interaction terms.
>
> Similar simulation methods have been widely used in studies involving clinical and behavioral data, with the primary goal of generating data that conform to a predefined Structural Causal Model. In this work, the covariate dimensionality in the simulation was relatively high. To increase the complexity of the relationship between covariates and the response variable, the final simulation employed a combination of linear and nonlinear terms.
>
> Subsequently, the process of averaging observations within the same time window and adding noise followed the methodology described in [2].

---

> ### Author Response · Authors · 2024-11-19
> **Weakness and Question: Strengths and weaknesses of our method across different conditions (including supplementary experiments on additional real-world datasets)**
>
> Before presenting the supplementary datasets and baselines, I will first summarize the strengths and weaknesses of our proposed method, as demonstrated through experiments and theoretical analysis. Following this, I will present comparative results on three additional datasets and four other longitudinal and time-series datasets using cutting-edge baselines, and analyze the reasons behind these results.
>
> **What kind of missing data imputation can our method handle well?**
>
> **1. MAR, MCAR, and high missing ratio:** Our experiments demonstrate that the missing mechanism in the ADNI dataset aligns with MAR (Missing At Random). Through logistic modeling and testing, we identified significant associations between the missing indicator of the response variable and key covariates such as Age and APOE4. This confirms that our method is well-suited for handling both MAR and MCAR conditions. However, we currently lack a theoretical explanation for its performance under MNAR (Missing Not At Random). The ADNI dataset also presents a challenging scenario, with a missing ratio for the response variable exceeding 90\%. Furthermore, in our simulations, we tested various missing ratios, and our method adapted well across these scenarios, demonstrating robustness under high missing ratios.
>
> **2. Block-wise missing:** Block-wise missing data, a common issue in biomedical datasets, poses unique challenges for data imputation. Our sub-sampling method, however, remains robust and unaffected by this issue. A clear instance of block-wise missing data can be observed in the ADNI dataset. Compared to other baseline methods, addressing block-wise missing data in mini-batches typically requires additional preprocessing. For instance, in VAE-based approaches, missing value imputation often involves initializing the missing values. This process becomes particularly challenging when block-wise missing data leads to significant gaps in the data indicators within a mini-batch. In contrast, Graph Neural Networks (GNNs) bypass the need for explicit initialization of missing values. Even in the presence of block-wise missing data within mini-batches, GNNs effectively utilize information from both corresponding observations of the same subject and observations from other subjects. This ability allows GNNs to learn robust and meaningful observation embeddings, ensuring their adaptability to such challenging scenarios.
>
> **Scenarios where our method may not be advantageous?**
>
> **1. Single time series and long-term memorized time series:** When faced with the imputation of missing data in a single-subject time series, the inductive learning capability of our proposed method across multiple subjects cannot be fully utilized. Unlike many cutting-edge RNN-based methods that focus on imputing single time series, our method is specifically designed for longitudinal data across multiple subjects, rather than simply filling gaps within the time series of an individual subject or system. Moreover, our model currently has limited ability to capture smooth, long-term temporal features in time series. RNN-based methods achieve this through memory and update components, concentrating all computational effort on the temporal axis while neglecting the relationships between multiple subjects. As a result, our method may not perform as well as RNNs in handling missing data for datasets with very long-term memory characteristics. Furthermore, our model currently has limited capacity to capture smooth, long-term temporal dependencies in time series. RNN-based methods excel in this domain due to their memory and update mechanisms, which focus computational effort exclusively on the temporal axis while neglecting inter-subject relationships. As a result, for datasets characterized by strong long-term memory features, RNN-based methods may outperform our approach in handling missing data.
>
> **2. Time series with weak feature correlations and high stability:** In datasets such as those used for electricity or air quality monitoring, the data are typically stable, well-aligned, and exhibit weak correlations across features. In such scenarios, RNN-based and VAE-based methods are already highly effective. The strengths of our model lie in its flexibility to handle irregular data, its inductive ability to capture interdependencies among features, and its capacity to learn specific representations for individual observations. Although our method performs competitively on time series with weak feature correlations and high stability, it does not offer a significant advantage over existing approaches in these cases.

---

> ### Author Response · Authors · 2024-11-19
> **Weakness and Question: Strengths and weaknesses of our method across different conditions (including supplementary experiments on additional real-world datasets)**
>
> The above summaries are drawn from both theoretical and experimental analyses. Furthermore, several time-series datasets with relevant characteristics were identified to evaluate the performance of our method in missing data imputation and prediction tasks. And we also add some other state-of-the-art baselines in longitudinal data/time series imputation for comparison.
> The supplementary baselines are:
>
> **[1]** Transformer: Ailing Zeng, Muxi Chen, Lei Zhang, and Qiang Xu. 2022. Are transformers effective for time series forecasting? arXiv preprint arXiv:2205.13504 (2022).
>
> **[2]** GP-VAE: Fortuin V, Baranchuk D, Rätsch G, et al. Gp-vae: Deep probabilistic time series imputation[C]//International conference on artificial intelligence and statistics. PMLR, 2020: 1651-1661.
>
> **[3]** SAITS: Du W, Côté D, Liu Y. Saits: Self-attention-based imputation for time series[J]. Expert Systems with Applications, 2023, 219: 119619.
>
> **[4]** CTA: Wi H, Shin Y, Park N. Continuous-time Autoencoders for Regular and Irregular Time Series Imputation[C]//Proceedings of the 17th ACM International Conference on Web Search and Data Mining. 2024: 826-835.
>
> For each dataset, we introduce missing data under an MCAR mechanism with a missing ratio of 0.3. We subsequently validated the imputation performance using the Mean Absolute Error (MAE) metric.
>
> |             | AirQuality | Electricity | Energy    |
> | ----------- | ---------- | ----------- | --------- |
> | Transformer | 0.220      | 0.889       | 0.313     |
> | GP-VAE      | 0.287      | 0.963       | 0.401     |
> | CTA         | **0.196**  | **0.797**   | 0.205     |
> | SAITS       | 0.207      | 0.894       | 0.301     |
> | GRAPE       | 0.251      | 0.874       | 0.211     |
> | IGRM        | 0.248      | 0.864       | 0.209     |
> | **SHT-GNN** | 0.203      | 0.814       | **0.183** |
>
> For each dataset, we also applied MCAR with a missing ratio of 0.5, and subsequently validated the missing data imputation results using MAE.
>
> |             | AirQuality | Electricity | Energy    |
> | ----------- | ---------- | ----------- | --------- |
> | Transformer | 0.235      | 0.946       | 0.393     |
> | GP-VAE      | 0.303      | 0.993       | 0.486     |
> | CTA         | **0.203**  | **0.812**   | 0.227     |
> | SAITS       | 0.220      | 0.943       | 0.351     |
> | GRAPE       | 0.269      | 0.892       | 0.239     |
> | IGRM        | 0.254      | 0.879       | 0.247     |
> | **SHT-GNN** | 0.211      | 0.823       | **0.203** |
>
> where the AirQuality dataset consists of a single-object time series with 15 features and over 9,000 records. The Electricity dataset comprises long-term time series data for 370 subjects, containing a total of over 140,256 records, each characterized by a single-dimensional attribute. The Energy dataset captures energy usage from various electrical appliances, representing 110 subjects with more than 10,000 observations and a total of 27 features.
>
> On the AirQuality and Electricity datasets, the SHT-GNN method demonstrates inferior performance compared to RNN and VAE-based approaches. This indicates that the SHT-GNN model is not well-suited for long-term, single-object time series, which fall outside its intended application scenario.
>
> However, SHT-GNN achieves state-of-the-art performance on the Energy dataset. Unlike the AirQuality and Electricity datasets, the Energy dataset involves time series with multiple subjects and multidimensional features. SHT-GNN leverages its ability to effectively borrow information across features, which significantly enhances its performance, particularly in the context of irregular longitudinal data imputation.

---

> ### Author Response · Authors · 2024-11-19
> **Questions: How to apply these tabular data/time series imputation methods in longitudinal data**
>
> **1. How to Apply Tabular Data Imputation to Longitudinal Data**
>
> For tabular data imputation, longitudinal data is concatenated observation-wise, regardless of the subject it originates from. All observations are separated and combined into a single tabular data structure, which is then used as the input for all missing data imputation methods.
>
> **2. How to Apply Time Series Imputation Methods to Longitudinal Data**
>
> In the tasks of missing data imputation and prediction for longitudinal data, two scenarios can be considered:
>
> For methods that provide code specifically designed for time series across multiple subjects, these can be directly applied to longitudinal data for imputation.
>
> For methods where the code only supports missing data imputation and prediction for a single time series, the imputation method will be applied separately to the observations corresponding to each subject.

---

> > ### Comment · Reviewer_mF3B · 2024-11-24
> >
> > Thank you for your clarification.
> >
> > The rebuttal from the authors primarily focuses on the following points:
> >
> > The theoretical intuition behind SHT-GNN can be understood from the perspective of VAE. However, it lacks a rigorous explanation, and I am concerned that this may not be the case. In VAE, the regularization term is computed based on the distance between latent space distributions. However, MADGap may not operate in the same manner.
> >
> >
> > Additionally, there are several key concerns that the authors have chosen to omit, such as W1.2, W1.3, W3.2, W3.3, and W3.4, which are critical to the argument.

---

> > > ### Author Response · Authors · 2024-11-24
> > > **Answer to W1.2**
> > >
> > > We developed a heterogeneous graph based on subject-wise sampling, diverging from the common GNN approaches of observation sampling in **mini-batch** training or **time-sliding window sampling** in RNNs. Our approach samples subjects from longitudinal data and forms heterogeneous graphs with their corresponding observations, which is **specifically tailored for longitudinal data scenarios**.

---

> > > ### Author Response · Authors · 2024-12-03
> > > **Answer to W1.3**
> > >
> > > Thank you for your suggestion! We have added the ablation study on real data. Specifically, we conducted the ablation study on the ADNI dataset.
> > >
> > > |             | Accuracy        | AUC             | RMSE            |
> > > | ----------- | --------------- | --------------- | --------------- |
> > > | None        | 0.714±0.009     | 0.724±0.010     | 0.106±0.002     |
> > > | One-Layer   | 0.733±0.013     | 0.764±0.014     | 0.102±0.002     |
> > > | Two-Layer   | **0.782±0.011** | **0.829±0.012** | **0.090±0.001** |
> > > | Three-Layer | 0.769±0.009     | 0.793±0.012     | 0.096±0.003     |
> > >
> > >
> > > As can be seen, the best performance is achieved with two layers. Therefore, the configuration we used for the final training and testing is based on two-layer longitudinal subnetwork.

---

> > > ### Author Response · Authors · 2024-12-03
> > > **Answer to W3.3 and W3.4**
> > >
> > > Thank you for your additional suggestions on the baseline! We have reviewed the most cutting-edge and already published work on imputation using longitudinal data.
> > >
> > > [1] CTA: Wi H, Shin Y, Park N. Continuous-time Autoencoders for Regular and Irregular Time Series Imputation[C]//Proceedings of the 17th ACM International Conference on Web Search and Data Mining. 2024: 826-835
> > >
> > > [2] SAITS: Du W, Côté D, Liu Y. Saits: Self-attention-based imputation for time series[J]. Expert Systems with Applications, 2023, 219: 119619.
> > >
> > > [3] GP-VAE: Fortuin V, Baranchuk D, Rätsch G, et al. Gp-vae: Deep probabilistic time series imputation[C]//International conference on artificial intelligence and statistics. PMLR, 2020: 1651-1661.
> > >
> > > [4] ReMasker: Imputing Tabular Data with Masked Autoencoding
> > >
> > >
> > >
> > > The different parameters we used on the ADNI and Synthetic data are due to the distinct temporal smoothing characteristics between them, mainly reflected in their varying degrees of temporal smoothing. Therefore, we applied longitudinal networks with different layers for temporal smoothing operations. To demonstrate the stability of our method across settings, we conducted experiments on an additional dataset using a unified configuration.
> > >
> > >
> > >
> > > **Configeration:**
> > >
> > > We train $\text{S\small{HT-GNN}}$ for 10 sampling phases with a sampling size of 200. For each sampled graph, we run 1500 training epochs using the Adam optimizer with a learning rate of 0.001. We employ a three-layer bipartite graph and three-layer longitudinal subnetworks for all subjects. We use the ReLU activation function as the non-linear activation function. The dimensions of both node embeddings and edge embeddings are set to 32. The message aggregation function $AGG_l$ is implemented as a mean pooling function $\text{MEAN(·)}$. Both $O_{impute}$ and $O_{predict}$ are implemented as multi-layer perceptrons (MLP) with 32 hidden units. The $\lambda$ in loss function is set to 0.001.
> > >
> > >
> > >
> > > **Experiment Result:**
> > >
> > > |                 | AirQuality      | Electricity     | Energy          | PhysioNet-2012  |
> > > | --------------- | --------------- | --------------- | --------------- | --------------- |
> > > | **MICE**        | 0.310±0.023     | 1.319±0.051     | 0.371±0.010     | 0.223±0.021     |
> > > | **3D MICE**     | 0.293±0.009     | 1.083±0.051     | 0.341±0.011     | 0.209±0.015     |
> > > | **GRAPE**       | 0.267±0.013     | 0.891±0.029     | 0.251±0.009     | 0.203±0.005     |
> > > | **IGRM**        | 0.242±0.010     | 0.867±0.045     | 0.231±0.013     | 0.193±0.014     |
> > > | **GP-VAE**      | 0.287±0.010     | 0.963±0.056     | 0.401±0.025     | 0.398±0.02      |
> > > | **Transformer** | 0.220±0.019     | 0.889±0.071     | 0.313±0.018     | 0.190±0.019     |
> > > | **SAITS**       | 0.201±0.009     | 0.894±0.051     | 0.301±0.012     | 0.190±0.014     |
> > > | **CATSI**       | 0.236±0.019     | 0.849±0.071     | 0.201±0.023     | 0.206±0.013     |
> > > | **Remask**      | 0.269±0.010     | 0.903±0.029     | 0.259±0.0013    | 0.213±0.007     |
> > > | **CTA**         | **0.196±0.012** | **0.767±0.042** | 0.205±0.019     | 0.192±0.016     |
> > > | **Our method**  | 0.212±0.010     | 0.834±0.025     | **0.183±0.011** | **0.187±0.009** |
> > >
> > > We have included the results of the above experiments in the revision. Thank you for your patient responses and valuable suggestions!

---

> ### Author Response · Authors · 2024-11-24
> **Answer to W3.2**
>
> The content you mentioned does not match what is stated in the GRAPE paper.
>
> In the original text of the GRAPE paper, it states: "For label prediction tasks, we use two GNN layers with 16 hidden units. $O_{edge}$ and $O_{node}$ are implemented as linear layers. The edge dropout rate is set to $r_{drop} = 0.3$. For all experiments, we run 5 trials with different random seeds and report the mean and standard deviation of the results."
>
> Clearly, the settings we are using here for the label prediction task in GRAPE are two GNN layers with 16 hidden units, not 64.

---

> > ### Comment · Reviewer_mF3B · 2024-11-24
> >
> > The GRAPE is capable of two tasks: 1) label prediction and 2) feature imputation
> >
> > Yes, when the task is label prediction, its hidden dimension is 16.
> >
> >
> > But, this paper studies the task of imputation, rather than the label prediction. As stated in Section 4.1 of the GRAPE paper, it is "For all feature imputation tasks, we use a 3-layer GNN with 64 hidden units and RELU activation."
> >
> > The topic that the paper evaluated is Imputation rather than label prediction.

---

> > > ### Author Response · Authors · 2024-11-24
> > >
> > > Thank for your feedback! Our paper focuses on the prediction of response variables under missing data conditions. All methods are aimed at serving the prediction of response variables. In GRAPE, it is clearly stated that the **end-to-end** prediction mode is significantly superior to the **impute and predict** mode. For tasks involving the prediction of response variables, we believe it is necessary to use GRAPE's end-to-end response prediction mode, rather than the mode that first performs data imputation followed by prediction.

---

### Official Review · Reviewer_bJUc · 2024-11-01

**Soundness:** 3
**Presentation:** 2
**Contribution:** 3
**Rating:** 6
**Confidence:** 4

**Summary:**

The authors propose the Sampling-guided Heterogeneous Graph Neural Network (SHT-GNN) approach that explores GNN structures to learn to impute longitudinal irregular sampled multivariate/modal data. The approach relies on GNNs for linking observations through time exploring a subjects minibatched sampling stategy for scalable inference with edge weights defined in terms of temporal smoothing. The latter defined using the product of an exponential decay in time, overlap in missing pattern and cosine similarity of the associated nodes embeddings. To prevent overfitting the Mean Average Distance Gap is used as regularization during training. The approach is contrasted simple to more advanced approaches for irregular data imputation including some existing GNN based imputation methodologies on a synthetic (based on simulated response variables from a real GLOBEM dataset) and real longitudinal dataset (ADNI) finding superior imputation performance than the compared baselines as well as improved computational efficiency when compared to GRAPE and iGRM.

**Strengths:**

The paper is well written , easy to follow, and with nice illustrations explaining the approach.

The methodology is sound and the included components including temporal smoothing and regularization for oversmoothing in GNN well motivated.

The approach have merits both in compute and ability to impute exploring temporal and multivariate/multimodal structure of data.

**Weaknesses:**

Whereas the approach is sound and the experimental comparison reasonable, it is somewhat limited. I.e. two datasets of which one is with simulated responses.

Furthermore, the literature on imputation of time-series data is vast and in this space, the paper covers only some of these works, see for instance:
https://paperswithcode.com/task/imputation

Arguably many of these listed procedures do not handle irregular sampled time-series, however, there are still many relevant methods and datasets for time-series imputation of irregular data that it would be highly interesting and relevant to see the performance of the current methodology against beyond the considered baselines. In particular the authors should consult Table 3 in the following recent survey of imputation of irregular time-series data for methods and datasets:
https://www.sciencedirect.com/science/article/pii/S0925231221003003
as well as the recently published article on imputation of irregular time-series data using autoencoders:
https://dl.acm.org/doi/10.1145/3616855.3635831

In particular, health care data such as MIMIC-III and the Physionet 2012 ICU challenge could here be relevant to consider as previously used and potentially also the other ICU datasets here listed.

Methodology-wise it would be interesting to compare against the recent autoencoder framework of:
https://dl.acm.org/doi/pdf/10.1145/3616855.3635831
as well as, BRITS, GP-VAE, SAITS also here compared against.

Whereas the approach is sound and I believe with merits the experimentation is currently too limited to fully see how meritable the approach is and its impact upon this very large body of existing literature.


Minor:
Dicision Tree -> Decision Tree

**Questions:**

Could the authors consider including more datasets, for instance as used previously for imputation of irregular data as referenced above? See in particular datasets and methodologies used here:
https://www.sciencedirect.com/science/article/pii/S0167947317300403
https://www.sciencedirect.com/science/article/pii/S0925231221003003
https://dl.acm.org/doi/10.1145/3616855.3635831
In particular, the second reference list many benchmark data sets (Table 3) as well as existing methodologies here used that are not currently compared against. Given the vast literature addressing imputation of irregular time-series data I think the paper needs to establish results much more extensively in terms of data and compared methods and the study is currently in its experimentation rather limited and in my eyes too limited. This makes it hard to judge the impact and utility of the approach compared to this rather large literature. I therefore strongly encourage the authors to include additional experimentation on well-established irregular temporal imputation datasets also considering additional well established methodologies for imputation in such irregular data (as given in the recent references above).

In summary, I think the authors’ methodology is sound and potentially indeed meritable and worth publication, but at this point I find this too unclear as the experimentation in terms of datasets and compared methodologies is too limited.

---

> ### Author Response · Authors · 2024-11-19
> **Weakness and Question: Strengths and weaknesses of our method across different conditions (including supplementary experiments on additional real-world datasets)**
>
> Thank you very much for your valuable feedback, particularly for pointing out the insufficiency in our baseline and dataset comparisons.
> Before presenting the supplementary datasets and baselines, I will first summarize the strengths and weaknesses of our proposed method, as demonstrated through experiments and theoretical analysis. Following this, I will present comparative results on three additional datasets and four other longitudinal and time-series datasets using cutting-edge baselines, and analyze the reasons behind these results.
>
> **What kind of missing data imputation can our method handle well?**
>
> **1. MAR, MCAR, and high missing ratio:** Our experiments demonstrate that the missing mechanism in the ADNI dataset aligns with MAR (Missing At Random). Through logistic modeling and testing, we identified significant associations between the missing indicator of the response variable and key covariates such as Age and APOE4. This confirms that our method is well-suited for handling both MAR and MCAR conditions. However, we currently lack a theoretical explanation for its performance under MNAR (Missing Not At Random). The ADNI dataset also presents a challenging scenario, with a missing ratio for the response variable exceeding 90\%. Furthermore, in our simulations, we tested various missing ratios, and our method adapted well across these scenarios, demonstrating robustness under high missing ratios.
>
> **2. Block-wise missing:** Block-wise missing data, a common issue in biomedical datasets, poses unique challenges for data imputation. Our sub-sampling method, however, remains robust and unaffected by this issue. A clear instance of block-wise missing data can be observed in the ADNI dataset. Compared to other baseline methods, addressing block-wise missing data in mini-batches typically requires additional preprocessing. For instance, in VAE-based approaches, missing value imputation often involves initializing the missing values. This process becomes particularly challenging when block-wise missing data leads to significant gaps in the data indicators within a mini-batch. In contrast, Graph Neural Networks (GNNs) bypass the need for explicit initialization of missing values. Even in the presence of block-wise missing data within mini-batches, GNNs effectively utilize information from both corresponding observations of the same subject and observations from other subjects. This ability allows GNNs to learn robust and meaningful observation embeddings, ensuring their adaptability to such challenging scenarios.
>
> **Scenarios where our method may not be advantageous?**
>
> **1. Single time series and long-term memorized time series:** When faced with the imputation of missing data in a single-subject time series, the inductive learning capability of our proposed method across multiple subjects cannot be fully utilized. Unlike many cutting-edge RNN-based methods that focus on imputing single time series, our method is specifically designed for longitudinal data across multiple subjects, rather than simply filling gaps within the time series of an individual subject or system. Moreover, our model currently has limited ability to capture smooth, long-term temporal features in time series. RNN-based methods achieve this through memory and update components, concentrating all computational effort on the temporal axis while neglecting the relationships between multiple subjects. As a result, our method may not perform as well as RNNs in handling missing data for datasets with very long-term memory characteristics. Furthermore, our model currently has limited capacity to capture smooth, long-term temporal dependencies in time series. RNN-based methods excel in this domain due to their memory and update mechanisms, which focus computational effort exclusively on the temporal axis while neglecting inter-subject relationships. As a result, for datasets characterized by strong long-term memory features, RNN-based methods may outperform our approach in handling missing data.
>
> **2. Time series with weak feature correlations and high stability:** In datasets such as those used for electricity or air quality monitoring, the data are typically stable, well-aligned, and exhibit weak correlations across features. In such scenarios, RNN-based and VAE-based methods are already highly effective. The strengths of our model lie in its flexibility to handle irregular data, its inductive ability to capture interdependencies among features, and its capacity to learn specific representations for individual observations. Although our method performs competitively on time series with weak feature correlations and high stability, it does not offer a significant advantage over existing approaches in these cases.

---

> ### Author Response · Authors · 2024-11-19
> **Weakness and Question: Strengths and weaknesses of our method across different conditions (including supplementary experiments on additional real-world datasets)**
>
> The above summaries are drawn from both theoretical and experimental analyses. Furthermore, several time-series datasets with relevant characteristics were identified to evaluate the performance of our method in missing data imputation and prediction tasks. And we also add some other state-of-the-art baselines in longitudinal data/time series imputation for comparison.
> The supplementary baselines are:
>
> **[1]** Transformer: Ailing Zeng, Muxi Chen, Lei Zhang, and Qiang Xu. 2022. Are transformers effective for time series forecasting? arXiv preprint arXiv:2205.13504 (2022).
>
> **[2]** GP-VAE: Fortuin V, Baranchuk D, Rätsch G, et al. Gp-vae: Deep probabilistic time series imputation[C]//International conference on artificial intelligence and statistics. PMLR, 2020: 1651-1661.
>
> **[3]** SAITS: Du W, Côté D, Liu Y. Saits: Self-attention-based imputation for time series[J]. Expert Systems with Applications, 2023, 219: 119619.
>
> **[4]** CTA: Wi H, Shin Y, Park N. Continuous-time Autoencoders for Regular and Irregular Time Series Imputation[C]//Proceedings of the 17th ACM International Conference on Web Search and Data Mining. 2024: 826-835.
>
> For each dataset, we introduce missing data under an MCAR mechanism with a missing ratio of 0.3. We subsequently validated the imputation performance using the Mean Absolute Error (MAE) metric.
>
> |             | AirQuality | Electricity | Energy    |
> | ----------- | ---------- | ----------- | --------- |
> | Transformer | 0.220      | 0.889       | 0.313     |
> | GP-VAE      | 0.287      | 0.963       | 0.401     |
> | CTA         | **0.196**  | **0.797**   | 0.205     |
> | SAITS       | 0.207      | 0.894       | 0.301     |
> | GRAPE       | 0.251      | 0.874       | 0.211     |
> | IGRM        | 0.248      | 0.864       | 0.209     |
> | **SHT-GNN** | 0.203      | 0.814       | **0.183** |
>
> For each dataset, we also applied MCAR with a missing ratio of 0.5, and subsequently validated the missing data imputation results using MAE.
>
> |             | AirQuality | Electricity | Energy    |
> | ----------- | ---------- | ----------- | --------- |
> | Transformer | 0.235      | 0.946       | 0.393     |
> | GP-VAE      | 0.303      | 0.993       | 0.486     |
> | CTA         | **0.203**  | **0.812**   | 0.227     |
> | SAITS       | 0.220      | 0.943       | 0.351     |
> | GRAPE       | 0.269      | 0.892       | 0.239     |
> | IGRM        | 0.254      | 0.879       | 0.247     |
> | **SHT-GNN** | 0.211      | 0.823       | **0.203** |
>
> where the AirQuality dataset consists of a single-object time series with 15 features and over 9,000 records. The Electricity dataset comprises long-term time series data for 370 subjects, containing a total of over 140,256 records, each characterized by a single-dimensional attribute. The Energy dataset captures energy usage from various electrical appliances, representing 110 subjects with more than 10,000 observations and a total of 27 features.
>
> On the AirQuality and Electricity datasets, the SHT-GNN method demonstrates inferior performance compared to RNN and VAE-based approaches. This indicates that the SHT-GNN model is not well-suited for long-term, single-object time series, which fall outside its intended application scenario.
>
> However, SHT-GNN achieves state-of-the-art performance on the Energy dataset. Unlike the AirQuality and Electricity datasets, the Energy dataset involves time series with multiple subjects and multidimensional features. SHT-GNN leverages its ability to effectively borrow information across features, which significantly enhances its performance, particularly in the context of irregular longitudinal data imputation.

---

> > ### Comment · Reviewer_bJUc · 2024-11-21
> > **Thanks for the additional experimentation can you provide error bars and argue why you did not select the datasets I suggested**
> >
> > Dear Authors,
> >
> > Thanks for your rebuttal and additional experimentation which is much appreciated.
> >
> > However, I would appreciate to see the performance also on the existing synthetic study and ADNI datasets in the main paper as well using these baselines.
> >
> > Furthermore, is there a reason why you did not also consider the health care data I pointed to such as MIMIC-III and the Physionet 2012 ICU challenge as well as the other ICU datasets?
> >
> > Finally, for these new baseline experiments I would appreciate to see error-bars on the results as also reported in the main paper to probe the uncertainties associated with the reported results.

---

> > > ### Author Response · Authors · 2024-11-23
> > > **Additional experimentation (Part 1)**
> > >
> > > Here, we present the performance of all the added baselines on the synthetic data:
> > >
> > > | Missing Ratio                | $r_X=0.3, r_Y=0.3$    | $r_X=0.3, r_Y=0.3$     | $r_X=0.3, r_Y=0.3$    | $r_X=0.3, r_Y=0.5$    | $r_X=0.3, r_Y=0.5$     | $r_X=0.3, r_Y=0.5$    |
> > > | ---------------------------- | --------------------- | ---------------------- | --------------------- | --------------------- | ---------------------- | --------------------- |
> > > | **Window Size and Variance** | **$w=3, \sigma=0.1$** | **$w=5, \sigma=0.15$** | **$w=7, \sigma=0.2$** | **$w=3, \sigma=0.1$** | **$w=5, \sigma=0.15$** | **$w=7, \sigma=0.2$** |
> > > | **Mean**                     | 0.693±0.009           | 0.826±0.011            | 0.936±0.011           | 0.820±0.012           | 0.833±0.012            | 1.051±0.016           |
> > > | **MICE**                     | 0.724±0.051           | 0.851±0.042            | 0.978±0.038           | 0.825±0.051           | 0.863±0.052            | 0.920±0.041           |
> > > | **3D MICE**                  | 0.689±0.031           | 0.753±0.037            | 0.847±0.029           | 0.741±0.035           | 0.785±0.037            | 0.883±0.021           |
> > > | **GRAPE**                    | 0.671±0.013           | 0.786±0.020            | 0.865±0.034           | 0.765±0.013           | 0.799±0.027            | 0.935±0.037           |
> > > | **CATSI**                    | 0.701±0.034           | 0.732±0.026            | 0.832±0.023           | 0.749±0.047           | 0.748±0.038            | 0.885±0.027           |
> > > | **IGRM**                     | 0.682±0.015           | 0.768±0.011            | 0.874±0.013           | 0.786±0.034           | 0.831±0.020            | 0.928±0.012           |
> > > | **GP-VAE**                   | 0.733±0.011           | 0.793±0.018            | 0.851±0.021           | 0.731±0.023           | 0.769±0.025            | 0.933±0.030           |
> > > | **Transformer**              | 0.611±0.022           | 0.691±0.023            | 0.791±0.013           | 0.678±0.013           | 0.718±0.019            | 0.873±0.021           |
> > > | **SAITS**                    | 0.609±0.019           | 0.729±0.019            | 0.801±0.025           | 0.683±0.018           | 0.723±0.023            | 0.925±0.030           |
> > > | **CTA**                      | 0.581±0.010           | 0.678±0.010            | 0.778±0.019           | 0.653±0.015           | 0.673±0.013            | 0.835±0.019           |
> > > | **Our Method**               | **0.552±0.011**       | **0.650±0.014**        | **0.759±0.018**       | **0.623±0.013**       | **0.653±0.018**        | **0.818±0.020**       |
> > >
> > > On the ADNI dataset, their performance is as follows:
> > >
> > > | **Method**      | **RMSE**        | **AUC**         | **Accuracy**    |
> > > | --------------- | --------------- | --------------- | --------------- |
> > > | **Mean**        | 0.112±0.002     | 0.671±0.009     | 0.682±0.008     |
> > > | **MICE**        | 0.109±0.004     | 0.717±0.021     | 0.701±0.021     |
> > > | **3D MICE**     | 0.103±0.004     | 0.731±0.017     | 0.721±0.018     |
> > > | **GRAPE**       | 0.106±0.002     | 0.724±0.010     | 0.714±0.011     |
> > > | **CATSI**       | 0.104±0.003     | 0.713±0.017     | 0.708±0.013     |
> > > | **IGRM**        | 0.105±0.002     | 0.721±0.013     | 0.713±0.009     |
> > > | **Transformer** | 0.110±0.005     | 0.735±0.021     | 0.721±0.020     |
> > > | **GP-VAE**      | 0.112±0.007     | 0.738±0.011     | 0.719±0.017     |
> > > | **CTA**         | 0.101±0.003     | 0.751±0.013     | 0.721±0.016     |
> > > | **SAITS**       | 0.107±0.002     | 0.747±0.010     | 0.725±0.013     |
> > > | **Our Method**  | **0.090±0.001** | **0.829±0.012** | **0.782±0.011** |

---

> > > ### Author Response · Authors · 2024-11-23
> > > **Additional experimentation (Part 2)**
> > >
> > > In addition, regarding the PhysioNet 2012 ICU dataset that you mentioned, as the application process for this dataset took some time, we completed experiments on this dataset later. I will now report the performance on several new datasets (including the baselines from the previous paper).
> > >
> > > |                 | AirQuality      | Electricity     | Energy          | PhysioNet-2012  |
> > > | --------------- | --------------- | --------------- | --------------- | --------------- |
> > > | **Transformer** | 0.220±0.019     | 0.889±0.071     | 0.313±0.018     | 0.190±0.019     |
> > > | **GP-VAE**      | 0.287±0.010     | 0.963±0.056     | 0.401±0.025     | 0.398±0.02      |
> > > | **CTA**         | **0.196±0.012** | **0.767±0.042** | 0.205±0.019     | 0.192±0.016     |
> > > | **SAITS**       | 0.201±0.009     | 0.894±0.051     | 0.301±0.012     | 0.190±0.014     |
> > > | **MICE**        | 0.310±0.023     | 1.319±0.051     | 0.371±0.010     | 0.223±0.021     |
> > > | **3D MICE**     | 0.293±0.009     | 1.083±0.051     | 0.341±0.011     | 0.209±0.015     |
> > > | **GRAPE**       | 0.267±0.013     | 0.891±0.029     | 0.251±0.009     | 0.203±0.005     |
> > > | **CATSI**       | 0.236±0.019     | 0.849±0.071     | 0.201±0.023     | 0.206±0.013     |
> > > | **IGRM**        | 0.242±0.010     | 0.867±0.045     | 0.231±0.013     | 0.193±0.014     |
> > > | **Our method**  | 0.212±0.010     | 0.834±0.025     | **0.183±0.011** | **0.187±0.009** |
> > >
> > >
> > >
> > > It can be observed that our method demonstrates a significant advantage on clinical longitudinal follow-up data, including the ADNI and PhysioNet datasets.
> > >
> > > Regarding the MIMIC-III dataset, since its application process takes a considerable amount of time, we have not yet obtained the necessary permissions. Additionally, this dataset requires a complex preprocessing procedure (including quality control, etc.). We regret that we are unable to report the results of our method on this dataset at this time.

---

> ### Author Response · Authors · 2024-11-26
> **Additional experimentation and the final revision**
>
> In the above responses, we have added the results of all supplementary experiments (including results on the PhysioNet 2012 ICU challenge dataset, results of all additional baselines on synthetic datasets and ADNI dataset, and error-bars of all these results), and we have uploaded the revision that includes all of the above supplementary results.
>
> Once again, thank you for your valuable feedback and advice!! And we hope our responses have addressed all of your concerns.

---

> > ### Comment · Reviewer_bJUc · 2024-11-27
> > **I thank the authors for their additional experimentation**
> >
> > I thank the authors for their additional experimentation which has helped to position the paper in relation to existing work and benchmarks. I am based on their revisions and additional experimentation inclined to raise my score to a borderline accept.

---

> > > ### Author Response · Authors · 2024-11-27
> > >
> > > Thank you for your valuable suggestions and assistance! The dataset you provided, as well as the additional baselines, have made the comparisons in our work significantly more reliable.
> > >
> > > Additionally, we noticed that the score your vote for our work has not yet been updated to "borderline accept" (6 points) in the system. Could you kindly confirm if any changes were made? We greatly appreciate your thorough review and patience!

---

### Official Review · Reviewer_iwjQ · 2024-11-02

**Soundness:** 3
**Presentation:** 3
**Contribution:** 3
**Rating:** 6
**Confidence:** 2

**Summary:**

The goal of this paper is to present a scalable imputation method for handling missing data in longitudinal studies. A sampling-guided heterogeneous graph neural network (SHT-GNN) is proposed and it is both scalable and capable of learning effectively from irregular and inconsistent longitudinal observations. The proposed method is evaluated against eight baseline methods on synthetic data generated from real data, and it outperforms all baseline methods.

Missing data is a common issue in longitudinal studies, making this an important problem to address. However, imputation methods for longitudinal covariates have been extensively studied, and a key method by Yao et al. (2005) was overlooked and not included in the comparison study. Although this method was designed for Gaussian data, it performs well for non-Gaussian data in practice. It would be valuable to see how this method compares to SHT-GNN.

The authors claim that their method can accommodate arbitrary missing data. However, this may be an overstatement, as it seems likely that their approach is effective only for data missing completely at random. If the method is indeed applicable to missing-at-random or informatively missing schemes, this should be clarified.

Reference:

Yao, Müller and Wang (2005). Functional data analysis for sparse longitudinal data.
Journal of the American Statistical Association.

**Strengths:**

The topic of imputation for longitudinal data is significant, and the proposed solution is appealing.

**Weaknesses:**

An important method was omitted from the comparison study.

**Questions:**

1. In what contexts are longitudinal observations inconsistent? Can your method handle noise in longitudinal measurements?

2. Does the temporal smoothing method work for longitudinal data with only a few measurements per subject?

---

> ### Author Response · Authors · 2024-11-19
> **Weakness 1: Consideration of an ignored method**
>
> We are grateful for the insightful feedback and constructive comments provided by the reviewers on our paper.
>
> It is noteworthy that Yao et al. (2005) conducted a functional principal component analysis (FPCA) for sparse and irregular data. In their concluding remarks, they proposed the consideration of a regular design in which many data points may be missing for certain subjects. However, our work differs fundamentally from theirs in both scope and focus. While Yao et al. (2005) emphasized statistical theory and FPCA for longitudinal and functional data, particularly addressing the estimation of mean and covariance functions, our work prioritizes prediction accuracy in large-scale, irregular longitudinal studies characterized by missing data in both covariate and response variables. Specifically,
>
> In functional data analysis, as described by Yao et al. (2005), observed sparse data are treated as samples from random curves that represent an underlying smooth and continuous trajectory (function) over time. The repeated observations of each subject are modeled as realizations of a continuous random function, with the primary goal being to recover and predict the entire smooth trajectory for each subject. In contrast, our approach focuses only on discrete observed time points, with particular interest in capturing trends at specific time points rather than reconstructing continuous trajectories.
>
> Our approach combines graph-based learning with strategies to address missing data imputation challenges, whereas their methodology is rooted in functional data analysis. In the work of Yao et al. (2005), sparse and irregular observations refer to scenarios where entire observations are missing within otherwise continuous and dense data. Once the functional data are estimated, they can be used to generate dense observations to address the absence of observations. However, this approach cannot handle cases where covariates and responses are partially missing within the observed data during estimation and prediction. In this paper, we aim to address this issue, specifically focusing on the problem of partial feature missingness within sparse observations. Our work emphasizes improving the prediction accuracy and reducing the variance of missing data in each individual observation.

---

> ### Author Response · Authors · 2024-11-19
> **Weakness 2 and Question 1: Clarify the conditions for our method and what is "inconsistent observation"**
>
> **1. Clarify the conditions for our method**
>
> Our method is capable of handling missing data under MCAR (Missing Completely at Random) and MAR (Missing at Random) assumptions, even in scenarios with a high missing ratio. Our experiments demonstrate that the missing mechanism in the ADNI dataset aligns with MAR (Missing At Random). Through logistic modeling and testing, we identified significant associations between the missing indicator of the response variable and key covariates such as Age and APOE4. This confirms that our method is well-suited for handling both MAR and MCAR conditions. However, we currently lack a theoretical explanation for its performance under MNAR (Missing Not At Random). The ADNI dataset also presents a challenging scenario, with a missing ratio for the response variable exceeding 90\%. Furthermore, in our simulations, we tested various missing ratios, and our method adapted well across these scenarios, demonstrating robustness under high missing ratios.
>
> **2. Longitudinal observations often become inconsistent due to the following circumstances:**
>
> **2.1. Missing Data at Specific Time Points:** Observations for certain variables may be unavailable at specific time points, resulting in incomplete datasets. This leads to misalignment across time and variables, as not all measurements are available for every subject at each observation time.
>
> **2.2. Absence of Entire Observations at Certain Time Steps:** Complete data points, including all variables for a specific subject at a given time, may be missing. This disrupts the regular observation schedule, converting a consistent timeline into an irregular one.
>
> **2.3. Variability in Observation Frequencies Between Subjects:** Subjects may have differing numbers of observations or data collected at varying intervals. For example, one subject might be observed weekly, while another is observed monthly, resulting in inconsistent observation schedules across the dataset.
>
> **2.4. Irregular Data Collection Schedules Due to External Factors:** Practical constraints, such as missed appointments or technical issues during data collection, can lead to observations being recorded at irregular intervals, further complicating the consistency of longitudinal data.
>
> In the last revision, we have emphasized these inconsistencies in the introduction to underscore their significance in longitudinal data analysis.

---

> > ### Comment · Reviewer_iwjQ · 2024-11-25
> >
> > Thanks for your clarification. I am satisfied with your response to weakness 1 and am convinced that your method can handle missing data under MCAR. However, I am still not fully convinced that it works under MAR. Theoretical support will be needed to justify its effectiveness under MAR.

---

> > > ### Author Response · Authors · 2024-11-27
> > > **Final revision**
> > >
> > > In the above responses, we clarify the applicability of our method in MCAR and MAR scenarios, and these details have been added in the final revision.
> > >
> > > Specifically, in the revision, we have added Section 5.4, which explains in which situations our method excels, what types of data it is applicable to, and how it adapts to different missing data types. To support these clarifications, we have also included the results of all additional baselines on the newly added datasets in the appendix. Finally, we have also included a theoretical understanding of the method in the appendix, which explains the applicability of our method to MCAR and MAR scenarios.
> > >
> > > Once again, we thank you for your patience and feedback! We hope that we have addressed all of your concerns.

---

> > > ### Author Response · Authors · 2024-12-02
> > >
> > > Thank you for your valuable reply and suggestions!
> > >
> > > We have addressed your questions in the above answers and revision, and made changes according to your requirements. We look forward to receiving your feedback!

---

> ### Author Response · Authors · 2024-11-19
> **Questions 2: Handle noise in longitudinal measurements and longitudinal data with only a few measurements**
>
> **1. Does the temporal smoothing method work for longitudinal data with only a few measurements per subject?**
>
> Our method is effective in handling scenarios with limited measurements per subject.
>
> **Methodology:** The foundation of our approach lies in the representation power of graph neural networks (GNNs), which serve as the foundation for both missing data imputation and response prediction. Each observation derives its embedding from the observed values associated with its attributed edges. Leveraging the strong expressive capabilities of GNNs, our temporal smoothing mechanism achieves robust performance without relying on a large number of measurements.
>
> **Experimental Evidence:** In the ADNI dataset, the number of measurements varies considerably across subjects, ranging from as few as 10–15 follow-ups to as many as 30–40 follow-ups. Our approach is specifically designed to accommodate such variability, demonstrating strong adaptability to the structure of the ADNI data. Meanwhile, in our simulation studies, the GLOBEM human behavior dataset features subjects with only 10–20 measurements. Despite this limitation, our method has consistently exhibited strong performance in processing such longitudinal data, further underscoring its robustness and capacity to handle sparse observations.
>
> **2. Can your method handle noise in longitudinal measurements?**
>
> Indeed, our method is capable of handling noise in longitudinal measurements and has demonstrated excellent performance on simulation datasets with noisy data. In particular, we introduced noise during the generation of response variables, as detailed in Section 5.2 of our paper, which provides a comprehensive description of the process used to incorporate noise into the simulations.

---

> ### Author Response · Authors · 2024-11-25
> **Theoretical  understandings and effectiveness under MAR**
>
> We are able to provide a complete theoretical understanding of our method and explain its applicability to MAR from a theoretical perspective. For all of the following content, we will include it in our revision:
>
> From an optimization perspective, the missing data imputation process in our method is conceptually similar to Variational Autoencoders (VAEs), where the goal of the reconstruction step is to minimize the reconstruction error as part of the Evidence Lower Bound (ELBO). As is widely recognized, the training objective of a standard VAE for missing data imputation is expressed as:
>
> $$
> X_{obs} \xrightarrow[\text{MLPs}]{\text{Encode}} Z \xrightarrow[\text{MLPs}]{\text{Decode}} \hat{X}_{obs}
> $$
>
> $$
> \text{Maximize:} \ \ \log{p(x_{obs})} = \int\log{ q(z|x_{obs})\cdot{\frac{p{(z,x_{obs})}}{q{(z|x_{obs})}}}dz} \geq \int q(z|x_{obs}) \log{\frac{p(x_{obs},z)}{q(z|x_{obs})}} dz
> $$
>
> $$
> \text{That is to maximize: } \ \ \ E_{q(z|x_{obs})}\log{p(x_{obs}|z)} - D_{KL}[q(z|x_{obs})\parallel p(z)]
> $$
>
> where $E_{q(z|x)}\log{p(x|z)}$ represents the **reconstruction loss**, and $D_{KL}[q(z|x)\parallel p(z)]$ is the **regularization term**.
> When maximizing $\log{p(x_{obs})}$, it is guaranteed that the estimated results for missing data will be consistent in both MCAR and MAR scenarios, a point that has been emphasized in many studies [1][2].
>
> [1] Mattei P A, Frellsen J. MIWAE: Deep generative modelling and imputation of incomplete data sets[C]//International conference on machine learning. PMLR, 2019: 4413-4423.
>
> [2] Collier M, Nazabal A, Williams C K I. VAEs in the presence of missing data[J]. arXiv preprint arXiv:2006.05301, 2020.
>
> In our proposed SHT-GNN, we are also theoretically optimizing $\log{p(x_{obs})}$. Specifically, the calculation and training process can be described as follows:
>
> $$
> \text{Maximize:} \ \ \log{p(X_{obs})} = \int\log{ q(Z^L_{O},Z^L_{F}|X_{obs})\cdot{\frac{p{(Z^L_{O},Z^L_{F},X_{obs})}}{q{(Z^L_O,Z^L_F|X_{obs})}}}dZ} \geq \int q(Z^L_O, Z^L_F|X_{obs}) \log{\frac{p(X_{obs},Z^L_O,Z^L_F)}{q(Z^L_O,Z^L_F|X_{obs})}} dZ
> $$
> $\text{That is to maximize}$: $E_{q(Z^L_{O},Z^L_F|X_{obs})}\log{p(X_{obs}|Z^L_{O},Z^L_{F})} - D_{\text{KL}}[p(Z^L_O,Z^L_F|X_{obs}) \parallel p(Z^L_O,Z^L_F)]$
>
> where $X_{obs}$ denotes the observed values, $Z_{O}^{\text{init}}$ and $Z_{F}^{\text{init}}$ represent the initial embedding matrices of all observation and feature nodes, respectively. $Z_{O}^{\text{L}}$ and $Z_{F}^{\text{L}}$ denote the embedding matrices of all observation and feature nodes after $L$ layers of forward computation in SHT-GNN. Subsequently, the training objective in SHT-GNN is expressed as:
>
> $$
> \text{Maximize:} \ \ \log{p(X_{obs})} = \int\log{ q(Z^L_{O},Z^L_{F}|X_{obs})\cdot{\frac{p{(Z^L_{O},Z^L_{F},X_{obs})}}{q{(Z^L_O,Z^L_F|X_{obs})}}}dZ} \geq \int q(Z^L_O, Z^L_F|X_{obs}) \log{\frac{p(X_{obs},Z^L_O,Z^L_F)}{q(Z^L_O,Z^L_F|X_{obs})}} dZ
> $$
>
> $$
> \text{That is to maximize:} \ \ E_{q(Z^L_{O},Z^L_F|X_{obs})}\log{p(X_{obs}|Z^L_{O},Z^L_{F})} - D_{KL}[p(Z^L_O,Z^L_F|X_{obs})\parallel p(Z^L_O,Z^L_F)]
> $$
>
> where $E_{q(Z_{O},Z_F|X_{obs})}\log{p(X_{obs}|Z^L_{O},Z^L_{F})}$ represents the **reconstruction loss**. Here, $E_{q(Z_{O},Z_F|X_{obs})}\log{p(X_{obs}|Z^L_{O},Z^L_{F})}$ is the **joint distribution** over all observations, which **differs from** the $E_{q(z|x_{obs})}\log{p(x_{obs}|z)}$ in VAE.
>
> Previously, in the case of VAE, their expectation is calculated on the sample level, typically using the MSE over the observed values of all samples to approximate the reconstruction loss. The specific form is:
>
> $\text{Loss} = \frac{1}{N} {\sum_{i=1}^N \sum_{j=1}^p {m_{ij}}\cdot(\hat{X_{ij}}-X_{ij})^2}$
>
> where $m_{ij}$ is the missing indicator for the $j$-th feature in the $i$-th observation.
>
> In contrast, in SHT-GNN, our reconstruction loss is in the form of a joint distribution, and it is not possible to estimate it by averaging over the samples. This is why we use edge dropout in missing data imputation, because directly using the loss over all observed edges will not provide an effective estimate of $E_{q(Z_{O},Z_F|X_{obs})}\log{p(X_{obs}|Z^L_{O},Z^L_{F})}$. Specifically, we estimate the overall reconstruction loss by randomly calculating the loss on some edges in each batch of different heterogeneous graphs.
>
> **In principle, the SHT-GNN and VAE-based methods share conceptual similarities. Furthermore, the reconstruction loss used in both methods, along with the maximized target $p(X_{obs})$, theoretically indicates that our method is also capable of handling missing data in the MAR scenario, in the same way as VAE-based methods.**

---

> ### Author Response · Authors · 2024-12-03
>
> Thank you for your valuable reply and suggestions!
>
> We have addressed your questions in the above answers and revision, and made changes according to your requirements. We look forward to receiving your feedback!

---

### Official Review · Reviewer_NuNS · 2024-11-05

**Soundness:** 2
**Presentation:** 2
**Contribution:** 2
**Rating:** 5
**Confidence:** 3

**Summary:**

This paper introduces a method to impute data in multivariate longitudinal data.

**Strengths:**

The paper focuses on a timely topic, dealing with longitudinal health data.  Moreover, they apply it to a real-world dataset of interest, ADNI. The developed a method that, at least in theory, overcomes limitations of existing tools.

**Weaknesses:**

There are 4-5 pages of methods.  Many steps were taken, each with its own few decisions.  It was quite difficult for me to ascertain which steps were important, and how much each step mattered. The ablation study did not help me much, I was not sure precisely whether this was on the real data or the synthetic data.

The paper only shows experiments in which the SH-GNN method outperforms everything else, for all possible parameter values of the simulation, and also the real data.  That is a bit fishy.  No method outperforms all other methods in all other scenarios.  Especially one that runs faster and takes less memory.  I understand that there are likely no 'benchmark data' for this kind of problem.  And yet, my interests when I read a paper are to get very clear when I *should* use a method, and when I should not.  I get no real insight into that from this manuscript.  The claim seems to be that I should use it when I have multi-subject longitudinal data, with cross-sectional and cross-temporal missingness.  But, there are certainly conditions on the distribution and missingness under which other things are better, both in theory and practice.  What are the conditions?

I find it a bit weird that decision tree was included, rather than random forest, or some other decision forest based approach, since forests are known to be far superior to trees.

**Questions:**

1. The method contains many steps, I did not follow them all, to be completely honest, there is a lot of new notation and terminology.  It would be great to simply the exposition of the method, if possible.

2. Of all the fancy things included in the method, including temporal smoothing, subject-wise mini-batch sampling, and concatening lots of stuff, experiments that illustrate which of those steps matters most, and how much, would be much more informative.

3. A simulation setting in which something trivial and straightforward outperforms your method, would be instructive.

---

> ### Author Response · Authors · 2024-11-19
> **Weakness 1 and Question 1: Insight and simple exposition of our method**
>
> We thank the Reviewer for the valuable comment. We acknowledge that our paper is structured in multiple steps, which may have caused some difficulty in understanding. To address this, we offer the following intuitive explanation to provide a clearer understanding of our method.
>
> In practice, the process within SHT-GNN can be understood as a form of data reconstruction. From an optimization perspective, this process is conceptually similar to Variational Autoencoders (VAEs), where the goal of the reconstruction step is to minimize the reconstruction error as part of the Evidence Lower Bound (ELBO). As is widely recognized, the training objective of a standard VAE is expressed as:
>
> $$
> X \xrightarrow[\text{MLPs}]{\text{Encode}} Z \xrightarrow[\text{MLPs}]{\text{Decode}} \hat{X}
> $$
>
> $$
> \text{Maximize:} \ \ \log{p(x)} = \int\log{ q(z|x)\cdot{\frac{p{(z,x)}}{q{(z|x)}}}dz} \geq \int q(z|x) \log{\frac{p(x,z)}{q(z|x)}}dz
> $$
>
> $$
> \text{That is to maximize:} \ \ E_{q(z|x)}\log{p(x|z)} - D_{\text{KL}}[q(z|x)\parallel p(z)]
> $$
>
> where $E_{q(z|x)}\log{p(x|z)}$ represents the $\textbf{reconstruction loss}$, and $D_{\text{KL}}[q(z|x)\parallel p(z)]$ is the $\textbf{regularization term}$. In our proposed SHT-GNN, the calculation and training process can be described as follows:
>
> $$
> X_{\text{obs}}, Z^{\text{init}}_{O}, Z_F^{\text{init}}  \xrightarrow[\huge \mathcal{G}]{\text{Message Passing, Embedding Update}} Z^L_O, Z^L_F \xrightarrow[\textbf{MLPs}]{\text{Edge-wise Prediction as Missing Data Imputation}} \hat{X}
> $$
>
> where $X_{\text{obs}}$ denotes the observed values, $Z_{O}^{\text{init}}$ and $Z_{F}^{\text{init}}$ represent the initial embedding matrices of all observation and feature nodes, respectively. $Z_{O}^{\text{L}}$ and $Z_{F}^{\text{L}}$ denote the embedding matrices of all observation and feature nodes after $L$ layers of forward computation in SHT-GNN. Subsequently, the training objective in SHT-GNN is expressed as:
>
> $$
> \text{Maximize:} \ \ \log{p(X_{obs})} = \int\log{ q(Z^L_{O},Z^L_{F}|X_{obs})\cdot{\frac{p{(Z^L_{O},Z^L_{F},X_{obs})}}{q{(Z^L_O,Z^L_F|X_{obs})}}}dZ} \geq \int q(Z^L_O, Z^L_F|X_{obs}) \log{\frac{p(X_{obs},Z^L_O,Z^L_F)}{q(Z^L_O,Z^L_F|X_{obs})}} dZ
> $$
> $\text{That is to maximize}$: $E_{q(Z^L_{O},Z^L_F|X_{obs})}\log{p(X_{obs}|Z^L_{O},Z^L_{F})} - D_{\text{KL}}[p(Z^L_O,Z^L_F|X_{obs}) \parallel p(Z^L_O,Z^L_F)]$
>
> where $E_{q(Z_{O},Z_F|X_{obs})}\log{p(X_{obs}|Z_{O}^L,Z_{F}^L)}$ represents the $\textbf{reconstruction loss}$.
>
> In principle, the SHT-GNN and VAE-based methods share conceptual similarities. However, the key distinctions between the SHT-GNN and VAE-based approaches in the handling of longitudinal data are as follows.
>
> $\textbf{1. Enhanced Representation Capability through Graph Neural Networks (GNNs)}$. In SHT-GNN, the encoding and decoding processes utilize graph neural networks (GNNs) instead of the multilayer perceptrons (MLPs) employed in standard VAEs. GNNs offer significantly stronger representation capabilities by leveraging the inherent data structure and facilitating temporal smoothing across non-independent observations, which is critical in longitudinal settings.
>
> $\textbf{2. Distinct Regularization Term for Non-Independent Observations}$. In VAE-based methods, the regularization term for the latent space distribution, $D_{\text{KL}}[q(z|x) \parallel p(z)]$, quantifies the independent distributions of individual observations. By contrast, SHT-GNN incorporates a KL divergence term, $D_{\text{KL}}[p(Z^L_O,Z^L_F|X_{obs}) \parallel p(Z^L_O,Z^L_F)]$, which measures the joint distribution of all observation representations. To account for dependencies within longitudinal data, SHT-GNN employs multi-layer temporal smoothing and introduces MADGap, enabling improved representation of the joint latent space distribution.
>
> In summary, the foundational intuition behind SHT-GNN lies in optimizing the reconstruction error during the data reconstruction process. Moreover, in the context of longitudinal data, SHT-GNN introduces temporal smoothing and MADGap to strike a balance between temporal consistency and the diversity of observation representations within the latent space.
>
> We have realized that the lack of straightforward presentation and theoretical intuitiveness in our paper has caused difficulties for readers. Therefore, in the revision of our paper, we will include all the aforementioned theoretical understandings as a remark within the article.

---

> > ### Comment · Reviewer_NuNS · 2024-11-24
> > **Novelty was unclear**
> >
> > Hi - I'm sorry my previous comment was unclear.  In the 4-5 pages of methods, what was unclear was the novelty.  What did you create? What did you apply? What did you modify? I could not figure that out.  I understand the basic mathematics of VAE-like approaches

---

> > > ### Author Response · Authors · 2024-11-24
> > > **Answer about the novelty**
> > >
> > > Thank you for your feedback! We will briefly summarize the issues related to novelty as follows:
> > >
> > > **What I have created:**
> > >
> > > 1. We **created** a multi-layer mechanism that accomplishes temporal smoothing between nodes via message passing and representation update, with **all weights and smoothing mechanisms**  being entirely novel;
> > > 2. We employed MADGap, which is a classic training objective in GNNs to enhance representation capability. We have originally adapted it for longitudinal data, **completely rewriting** its form to better trade off between temporal smoothing and representation specificity.
> > > 3. We **created** a heterogeneous graph based on subject-wise sampling, diverging from the common machine learning approaches of observation sampling in mini-batch training or time-sliding window sampling in RNNs. Our approach samples subjects from longitudinal data and forms heterogeneous graphs with their corresponding observations.
> > >
> > > **What I have applied and modified:**
> > >
> > > 1. We **utilized** the classic GraphSAGE mechanism on a bipartite graph, while also **modifying** it by adding new components like edge embeddings and introducing an edge update step. This is done to perform inductive learning on observed data, making it more suitable for our current problem context of missing data imputation.
> > > 2. We **applied** the parameter-sharing capabilities of graph neural networks to the heterogeneous graphs generated by our subject-wise sampling approach.

---

> > > ### Author Response · Authors · 2024-11-30
> > > **Final Revision**
> > >
> > > In the above responses, we clarify the novelty of our work (What did we create? What did we apply? What did we modify?). We have uploaded the revision that includes all of these clarifications. Specifically, we have **added Section 5.4**, which explains in which situations our method excels, what types of data it is applicable to, and how it adapts to different missing data types.
> > >
> > > To support these clarifications, we have also included the results of all additional baselines on the newly added datasets in the **added appendix**. In the appendix, we present the experiments **using graphics** as you suggested, rather than large tables. Finally, we have included a theoretical understanding of the method in the appendix as well.
> > >
> > > Once again, we thank you for your patience and feedback!! We hope that we have addressed all of your concerns.

---

> > > ### Author Response · Authors · 2024-12-02
> > >
> > > Thank you for your valuable reply and suggestions!
> > >
> > > We have addressed your questions in the above answers and revision, and made changes according to your requirements. We look forward to receiving your feedback!

---

> ### Author Response · Authors · 2024-11-19
> **Weakness 2 and Question 2: Identifying key modules in our method**
>
> Thank you for your valuable feedback! I will clarify which modules in our proposed method contribute the most to the performance of our model:
>
> $\textbf{1. The inductive learning and representation capability of GNN as the framework’s foundation}$: Graph Neural Networks (GNNs) provide the foundational strength of this framework, particularly through their ability to effectively handle longitudinal data with irregular and inconsistent observation schedules. Unlike RNN-based methods, GNNs demonstrate superior representation capabilities, which underpin the accuracy of SHT-GNN in both missing data imputation and longitudinal data prediction. At each computational step, operations such as concatenation, aggregation, and updates adhere to the principles of GNNs, enabling the effective learning of observation embeddings. This robust representation capability is crucial for capturing complex relationships within irregular longitudinal datasets, ensuring the framework’s strong performance.
>
> $\textbf{2. Temporal smoothing mechanism and MADGap as key components for data reconstruction in longitudinal data}$: The superior performance of this method arises from the multi-layer temporal smoothing mechanism, specifically designed for longitudinal data, and the incorporation of MADGap in the loss function. This combination facilitates a trainable trade-off between temporal smoothing and the preservation of the unique characteristics of observation representations, which is critical for effective data reconstruction in longitudinal datasets.
>
> $\textbf{3. Mini-batch subject-wise sampling is not critical to the performance of SHT-GNN}$. The primary role of mini-batch subject-wise sampling is to address the memory storage challenges posed by massive datasets through the use of sharable trainable parameters. While it does not directly enhance the model’s accuracy or generalization capabilities, it demonstrates the feasibility of distributed training for the proposed model on large-scale datasets. For instance, this approach enables parallel training on multiple subgraphs using a gradient-sharing strategy, ensuring scalability for large-scale implementations.
> }

---

> > ### Author Response · Authors · 2024-11-19
> > **Weakness 3 and Question 3 (Part 1): Strengths and weaknesses of our method across different conditions (including supplementary experiments on additional real-world datasets)**
> >
> > **What types of missing data imputation can our method handle effectively?**
> >
> > **1. MAR, MCAR, and high missing ratio**:
> >
> > Our experiments demonstrate that the missing mechanism in the ADNI dataset aligns with MAR (Missing At Random). Through logistic modeling and testing, we identified significant associations between the missing indicator of the response variable and key covariates such as Age and APOE4. This confirms that our method is well-suited for handling both MAR and MCAR conditions. From a theoretical perspective, under the simplified data reconstruction framework described earlier, our proposed model can be viewed as reconstructing $p(X_{\text{obs}})$ under MAR and MCAR (It has been proven in "Handling Incomplete Data with Variational Autoencoders" that VAE can accommodate MAR, which can naturally be extended to our method as well). However, we currently lack a theoretical explanation for its performance under MNAR (Missing Not At Random). The ADNI dataset also presents a challenging scenario, with a missing ratio for the response variable exceeding 90\%. Furthermore, in our simulations, we tested various missing ratios, and our method adapted well across these scenarios, demonstrating robustness under high missing ratios.
> >
> > **2. Block-wise missing:**
> >
> > Block-wise missing data, a common issue in biomedical datasets, poses unique challenges for data imputation. Our sub-sampling method, however, remains robust and unaffected by this issue. A clear instance of block-wise missing data can be observed in the ADNI dataset. Compared to other baseline methods, addressing block-wise missing data in mini-batches typically requires additional preprocessing. For instance, in VAE-based approaches, missing value imputation often involves initializing the missing values. This process becomes particularly challenging when block-wise missing data leads to significant gaps in the data indicators within a mini-batch. In contrast, Graph Neural Networks (GNNs) bypass the need for explicit initialization of missing values. Even in the presence of block-wise missing data within mini-batches, GNNs effectively utilize information from both corresponding observations of the same subject and observations from other subjects. This ability allows GNNs to learn robust and meaningful observation embeddings, ensuring their adaptability to such challenging scenarios.
> >
> > **Scenarios where our method may not be advantageous?**
> >
> > **1. Single time series and long-term memorized time series:**
> >
> > When faced with the imputation of missing data in a single-subject time series, the inductive learning capability of our proposed method across multiple subjects cannot be fully utilized. Unlike many cutting-edge RNN-based methods that focus on imputing single time series, our method is specifically designed for longitudinal data across multiple subjects, rather than simply filling gaps within the time series of an individual subject or system. Moreover, our model currently has limited ability to capture smooth, long-term temporal features in time series. RNN-based methods achieve this through memory and update components, concentrating all computational effort on the temporal axis while neglecting the relationships between multiple subjects. As a result, our method may not perform as well as RNNs in handling missing data for datasets with very long-term memory characteristics. Furthermore, our model currently has limited capacity to capture smooth, long-term temporal dependencies in time series. RNN-based methods excel in this domain due to their memory and update mechanisms, which focus computational effort exclusively on the temporal axis while neglecting inter-subject relationships. As a result, for datasets characterized by strong long-term memory features, RNN-based methods may outperform our approach in handling missing data.
> >
> > **2. Time series with weak feature correlations and high stability:**
> >
> > In datasets such as those used for electricity or air quality monitoring, the data are typically stable, well-aligned, and exhibit weak correlations across features. In such scenarios, RNN-based and VAE-based methods are already highly effective. The strengths of our model lie in its flexibility to handle irregular data, its inductive ability to capture interdependencies among features, and its capacity to learn specific representations for individual observations. Although our method performs competitively on time series with weak feature correlations and high stability, it does not offer a significant advantage over existing approaches in these cases.

---

> ### Author Response · Authors · 2024-11-19
> **Weakness 3 and Question 3 (Part 2): Strengths and weaknesses of our method across different conditions (including supplementary experiments on additional real-world datasets)**
>
> The above summaries are drawn from both theoretical and experimental analyses. Furthermore, several time-series datasets with relevant characteristics were identified to evaluate the performance of our method in missing data imputation and prediction tasks. And we also add some other state-of-the-art baselines in longitudinal data/time series imputation for comparison.
> The supplementary baselines are:
>
> **[1]** Transformer: Ailing Zeng, Muxi Chen, Lei Zhang, and Qiang Xu. 2022. Are transformers effective for time series forecasting? arXiv preprint arXiv:2205.13504 (2022).
>
> **[2]** GP-VAE: Fortuin V, Baranchuk D, Rätsch G, et al. Gp-vae: Deep probabilistic time series imputation[C]//International conference on artificial intelligence and statistics. PMLR, 2020: 1651-1661.
>
> **[3]** SAITS: Du W, Côté D, Liu Y. Saits: Self-attention-based imputation for time series[J]. Expert Systems with Applications, 2023, 219: 119619.
>
> **[4]** CTA: Wi H, Shin Y, Park N. Continuous-time Autoencoders for Regular and Irregular Time Series Imputation[C]//Proceedings of the 17th ACM International Conference on Web Search and Data Mining. 2024: 826-835.
>
> For each dataset, we introduce missing data under an MCAR mechanism with a missing ratio of 0.3. We subsequently validated the imputation performance using the Mean Absolute Error (MAE) metric.
>
> |             | AirQuality | Electricity | Energy    |
> | ----------- | ---------- | ----------- | --------- |
> | Transformer | 0.220      | 0.889       | 0.313     |
> | GP-VAE      | 0.287      | 0.963       | 0.401     |
> | CTA         | **0.196**  | **0.797**   | 0.205     |
> | SAITS       | 0.207      | 0.894       | 0.301     |
> | GRAPE       | 0.251      | 0.874       | 0.211     |
> | IGRM        | 0.248      | 0.864       | 0.209     |
> | **SHT-GNN** | 0.203      | 0.814       | **0.183** |
>
> For each dataset, we also applied MCAR with a missing ratio of 0.5, and subsequently validated the missing data imputation results using MAE.
>
> |             | AirQuality | Electricity | Energy    |
> | ----------- | ---------- | ----------- | --------- |
> | Transformer | 0.235      | 0.946       | 0.393     |
> | GP-VAE      | 0.303      | 0.993       | 0.486     |
> | CTA         | **0.203**  | **0.812**   | 0.227     |
> | SAITS       | 0.220      | 0.943       | 0.351     |
> | GRAPE       | 0.269      | 0.892       | 0.239     |
> | IGRM        | 0.254      | 0.879       | 0.247     |
> | **SHT-GNN** | 0.211      | 0.823       | **0.203** |
>
> where the AirQuality dataset consists of a single-object time series with 15 features and over 9,000 records. The Electricity dataset comprises long-term time series data for 370 subjects, containing a total of over 140,256 records, each characterized by a single-dimensional attribute. The Energy dataset captures energy usage from various electrical appliances, representing 110 subjects with more than 10,000 observations and a total of 27 features.
>
> On the AirQuality and Electricity datasets, the SHT-GNN method demonstrates inferior performance compared to RNN and VAE-based approaches. This indicates that the SHT-GNN model is not well-suited for long-term, single-object time series, which fall outside its intended application scenario.
>
> However, SHT-GNN achieves state-of-the-art performance on the Energy dataset. Unlike the AirQuality and Electricity datasets, the Energy dataset involves time series with multiple subjects and multidimensional features. SHT-GNN leverages its ability to effectively borrow information across features, which significantly enhances its performance, particularly in the context of irregular longitudinal data imputation.

---

> ### Author Response · Authors · 2024-11-23
> **Additional experimentation (Part 1)**
>
> Here, we present the performance of all the added baselines on the synthetic data:
>
> | Missing Ratio                | $r_X=0.3, r_Y=0.3$    | $r_X=0.3, r_Y=0.3$     | $r_X=0.3, r_Y=0.3$    | $r_X=0.3, r_Y=0.5$    | $r_X=0.3, r_Y=0.5$     | $r_X=0.3, r_Y=0.5$    |
> | ---------------------------- | --------------------- | ---------------------- | --------------------- | --------------------- | ---------------------- | --------------------- |
> | **Window Size and Variance** | **$w=3, \sigma=0.1$** | **$w=5, \sigma=0.15$** | **$w=7, \sigma=0.2$** | **$w=3, \sigma=0.1$** | **$w=5, \sigma=0.15$** | **$w=7, \sigma=0.2$** |
> | **Mean**                     | 0.693±0.009           | 0.826±0.011            | 0.936±0.011           | 0.820±0.012           | 0.833±0.012            | 1.051±0.016           |
> | **MICE**                     | 0.724±0.051           | 0.851±0.042            | 0.978±0.038           | 0.825±0.051           | 0.863±0.052            | 0.920±0.041           |
> | **3D MICE**                  | 0.689±0.031           | 0.753±0.037            | 0.847±0.029           | 0.741±0.035           | 0.785±0.037            | 0.883±0.021           |
> | **GRAPE**                    | 0.671±0.013           | 0.786±0.020            | 0.865±0.034           | 0.765±0.013           | 0.799±0.027            | 0.935±0.037           |
> | **CATSI**                    | 0.701±0.034           | 0.732±0.026            | 0.832±0.023           | 0.749±0.047           | 0.748±0.038            | 0.885±0.027           |
> | **IGRM**                     | 0.682±0.015           | 0.768±0.011            | 0.874±0.013           | 0.786±0.034           | 0.831±0.020            | 0.928±0.012           |
> | **GP-VAE**                   | 0.733±0.011           | 0.793±0.018            | 0.851±0.021           | 0.731±0.023           | 0.769±0.025            | 0.933±0.030           |
> | **Transformer**              | 0.611±0.022           | 0.691±0.023            | 0.791±0.013           | 0.678±0.013           | 0.718±0.019            | 0.873±0.021           |
> | **SAITS**                    | 0.609±0.019           | 0.729±0.019            | 0.801±0.025           | 0.683±0.018           | 0.723±0.023            | 0.925±0.030           |
> | **CTA**                      | 0.581±0.010           | 0.678±0.010            | 0.778±0.019           | 0.653±0.015           | 0.673±0.013            | 0.835±0.019           |
> | **Our Method**               | **0.552±0.011**       | **0.650±0.014**        | **0.759±0.018**       | **0.623±0.013**       | **0.653±0.018**        | **0.818±0.020**       |
>
>
>
> On the ADNI dataset, their performance is as follows:
>
> | **Method**      | **RMSE**        | **AUC**         | **Accuracy**    |
> | --------------- | --------------- | --------------- | --------------- |
> | **Mean**        | 0.112±0.002     | 0.671±0.009     | 0.682±0.008     |
> | **MICE**        | 0.109±0.004     | 0.717±0.021     | 0.701±0.021     |
> | **3D MICE**     | 0.103±0.004     | 0.731±0.017     | 0.721±0.018     |
> | **GRAPE**       | 0.106±0.002     | 0.724±0.010     | 0.714±0.011     |
> | **CATSI**       | 0.104±0.003     | 0.713±0.017     | 0.708±0.013     |
> | **IGRM**        | 0.105±0.002     | 0.721±0.013     | 0.713±0.009     |
> | **Transformer** | 0.110±0.005     | 0.735±0.021     | 0.721±0.020     |
> | **GP-VAE**      | 0.112±0.007     | 0.738±0.011     | 0.719±0.017     |
> | **CTA**         | 0.101±0.003     | 0.751±0.013     | 0.721±0.016     |
> | **SAITS**       | 0.107±0.002     | 0.747±0.010     | 0.725±0.013     |
> | **Our Method**  | **0.090±0.001** | **0.829±0.012** | **0.782±0.011** |

---

> ### Author Response · Authors · 2024-11-23
> **Additional experimentation (Part 2)**
>
> In addition, regarding the PhysioNet 2012 ICU dataset, we also completed experiments on this dataset.
>
> |                 | AirQuality      | Electricity     | Energy          | PhysioNet-2012  |
> | --------------- | --------------- | --------------- | --------------- | --------------- |
> | **Transformer** | 0.220±0.019     | 0.889±0.071     | 0.313±0.018     | 0.190±0.019     |
> | **GP-VAE**      | 0.287±0.010     | 0.963±0.056     | 0.401±0.025     | 0.398±0.02      |
> | **CTA**         | **0.196±0.012** | **0.767±0.042** | 0.205±0.019     | 0.192±0.016     |
> | **SAITS**       | 0.201±0.009     | 0.894±0.051     | 0.301±0.012     | 0.190±0.014     |
> | **MICE**        | 0.310±0.023     | 1.319±0.051     | 0.371±0.010     | 0.223±0.021     |
> | **3D MICE**     | 0.293±0.009     | 1.083±0.051     | 0.341±0.011     | 0.209±0.015     |
> | **GRAPE**       | 0.267±0.013     | 0.891±0.029     | 0.251±0.009     | 0.203±0.005     |
> | **CATSI**       | 0.236±0.019     | 0.849±0.071     | 0.201±0.023     | 0.206±0.013     |
> | **IGRM**        | 0.242±0.010     | 0.867±0.045     | 0.231±0.013     | 0.193±0.014     |
> | **Our method**  | 0.212±0.010     | 0.834±0.025     | **0.183±0.011** | **0.187±0.009** |
>
> It can be observed that our method demonstrates a significant advantage on clinical longitudinal follow-up data, including the ADNI and PhysioNet datasets.

---

> ### Comment · Reviewer_NuNS · 2024-11-24
> **How will you revise?**
>
> Seems like you have added some additional experiments to address my other comments.  However, it is unclear to me based on the additional experiments if you have addressed these:
>
> 1. Varying degrees of those types of missingness, to empirically evaluate against other methods?
> 2. Single-time series and weak feature correlations, to show where the methods break down?
>
> Also, in general, I find large tables incomprehensible, and prefer graphics. See various blog posts from Andrew Gelman on the topic.

---

> ### Author Response · Authors · 2024-11-24
>
> **Concerning the missing mechanism:**
>
> In both the ADNI dataset and the PhysioNet 2012 ICU dataset, there is a significant presence of a MAR (Missing At Random) mechanism:
>
> 1. Existing studies indicate that the missing data in these datasets follows a MAR mechanism. The strong performance of our methods in these datasets reflects our approach's capability to effectively handle MAR mechanisms.
>
>     **[1]** Campos S, Pizarro L, Valle C, et al. Evaluating imputation techniques for missing data in ADNI: a patient classification study[C]//Progress in Pattern Recognition, Image Analysis, Computer Vision, and Applications: 20th Iberoamerican Congress, CIARP 2015, Montevideo, Uruguay, November 9-12, 2015, Proceedings 20. Springer International Publishing, 2015: 3-10.
>
>     **[2]** Lo R Y, Jagust W J, Aisen P, et al. Predicting missing biomarker data in a longitudinal study of Alzheimer disease[J]. Neurology, 2012, 78(18): 1376-1382.
>
> 2. Regarding experiments under the MNAR (Missing Not At Random) perspective, we will add these if time permits.
>
> **Single time series and weak feature correlation:**
>
> 1. Single time series: The AirQuality dataset consists of a single object time series with over 9,000 records and 15 features.
>
> 2. Weak feature correlation: The Electricity dataset contains long-term time series data for 370 locations, totaling over 140,256 records, with each record including data collected at 370 locations at the current time point. There is weak correlation between different locations (i.e., different features).
>
> From the results presented above, it is evident that our method does not perform notably on the AirQuality and Electricity datasets. Regarding the consideration of new baselines, we have added many cutting-edge baselines for longitudinal data imputation from the years 2022 to 2024, specifically including:
>
> **[1]** Transformer: Ailing Zeng, Muxi Chen, Lei Zhang, and Qiang Xu. 2022. Are transformers effective for time series forecasting? arXiv preprint arXiv:2205.13504 (2022).
>
> **[2]** GP-VAE: Fortuin V, Baranchuk D, Rätsch G, et al. Gp-vae: Deep probabilistic time series imputation[C]//International conference on artificial intelligence and statistics. PMLR, 2020: 1651-1661.
>
> **[3]** SAITS: Du W, Côté D, Liu Y. Saits: Self-attention-based imputation for time series[J]. Expert Systems with Applications, 2023, 219: 119619.
>
> **[4]** CTA: Wi H, Shin Y, Park N. Continuous-time Autoencoders for Regular and Irregular Time Series Imputation[C]//Proceedings of the 17th ACM International Conference on Web Search and Data Mining. 2024: 826-835.
>
>  **Problems about large tables**
>
> Finally, regarding the final presentation style, thank you for your suggestion! We will change the large tables to graphical form in the final revision.

---

> ### Author Response · Authors · 2024-11-27
> **Final revision**
>
> In the above responses, we clarify the novelty of our work (What did we create? What did we apply? What did we modify?). We have uploaded the revision that includes all of these clarifications. Specifically, we have **added Section 5.4**, which explains in which situations our method excels, what types of data it is applicable to, and how it adapts to different missing data types.
>
> To support these clarifications, we have also included the results of all additional baselines on the newly added datasets in the **added appendix**. In the appendix, we present the experiments **using graphics** as you suggested, rather than large tables. Finally, we have included a theoretical understanding of the method in the appendix as well.
>
> Once again, we thank you for your patience and feedback!! We hope that we have addressed all of your concerns.

---

> ### Author Response · Authors · 2024-12-03
>
> Thank you for your valuable reply and suggestions!
>
> We have addressed your questions in the above answers and revision, and made changes according to your requirements. We look forward to receiving your feedback!

---

### Official Review · Reviewer_iXB5 · 2024-11-06

**Soundness:** 3
**Presentation:** 3
**Contribution:** 3
**Rating:** 6
**Confidence:** 3

**Summary:**

The paper proposes Sampling-guided Heterogeneous Graph Neural Network (SHT-GNN) for Longitudinal data imputation. SHT-GNN treats both observations and covariates as nodes, and connects observations at successive time points, while keep covariate-observation interactions in a bipartite graph. Three key component in this model: subject-wise mini-batch sampling to reduce computation burden, inductive learning from observed edges and temporal smoothing in longitudinal subnetworks. With the embeddings generated with GNN, the authors employs MLPs to get the imputed covariates and predict the response. The loss function also takes consideration of a term Mean Average Distance Gap to prevent over-smoothing of GNN. Extensive experiment results and ablation studies are provided with semi-simulated and real data to demonstrate the effectiveness of the proposed framework.

**Strengths:**

- The paper is well written and easy to follow. Illustrations make it clear to understand the framework.
- The proposed framework, especially the way to construct the graph is novel.
- Experiments show promising results on the provided framework, especially on the real ADNI dataset, which could have significant impact on biomedical science.

**Weaknesses:**

- The paper is mainly heuristic on proposing a framework and show experiment results, without theoretical understanding of why it works. Although I do not think this is a deal breaker, it would be nice if the authors could provide more intuitions on why they design certain module the way it is. For example, it is unclear whether the way to get the weight with equation (7) is the best option, are there any alternative ways? How stable it is in practice and how to set $\gamma$ in practice?
- For covariate imputation, it seems to me one important baseline is missing, which is to do feature propagation with the proposed graph, similar to the way label propagation works. A recent work is [1]

[1] Rossi, Emanuele, et al. "On the unreasonable effectiveness of feature propagation in learning on graphs with missing node features." Learning on graphs conference. PMLR, 2022.

**Questions:**

- Is there any transformer based model for longitudinal data imputation? If there is work with sequence models like RNN and LSTM, I would think there should be transformer based models too. If so, they should also be baselines, and they could potentially mitigate the computation burden by parallel computing
- The response simulation model in Appendix 1 seems strange. Are you just combining different linear/non-linear functions together to make it complex or is there any intuition or real world model to guide the design?
- One suggestion is to change the abbreviation to SH-GNN

---

> ### Author Response · Authors · 2024-11-19
> **Weakness 1: Simplified intuition and theoretical understanding of our method**
>
> We thank the Reviewer for the insightful comment. In practice, the process within SHT-GNN can be understood as a form of data reconstruction. From an optimization perspective, this process is conceptually similar to Variational Autoencoders (VAEs), where the goal of the reconstruction step is to minimize the reconstruction error as part of the Evidence Lower Bound (ELBO). As is widely recognized, the training objective of a standard VAE is expressed as:
>
> $$
> X \xrightarrow[\text{MLPs}]{\text{Encode}} Z \xrightarrow[\text{MLPs}]{\text{Decode}} \hat{X}
> $$
>
> $$
> \text{Maximize:} \ \ \log{p(x)} = \int\log{ q(z|x)\cdot{\frac{p{(z,x)}}{q{(z|x)}}}dz} \geq \int q(z|x) \log{\frac{p(x,z)}{q(z|x)}}dz
> $$
>
> $$
> \text{That is to maximize:} \ \ E_{q(z|x)}\log{p(x|z)} - D_{\text{KL}}[q(z|x)\parallel p(z)]
> $$
> where $E_{q(z|x)}\log{p(x|z)}$ represents the $\textbf{reconstruction loss}$, and $D_{\text{KL}}[q(z|x)\parallel p(z)]$ is the $\textbf{regularization term}$. In our proposed SHT-GNN, the calculation and training process can be described as follows:
>
> $$
> X_{\text{obs}}, Z^{\text{init}}_{O}, Z_F^{\text{init}}  \xrightarrow[\huge \mathcal{G}]{\text{Message Passing, Embedding Update}} Z^L_O, Z^L_F \xrightarrow[\textbf{MLPs}]{\text{Edge-wise Prediction as Missing Data Imputation}} \hat{X}
> $$
>
> where $X_{\text{obs}}$ denotes the observed values, $Z_{O}^{\text{init}}$ and $Z_{F}^{\text{init}}$ represent the initial embedding matrices of all observation and feature nodes, respectively. $Z_{O}^{\text{L}}$ and $Z_{F}^{\text{L}}$ denote the embedding matrices of all observation and feature nodes after $L$ layers of forward computation in SHT-GNN. Subsequently, the training objective in SHT-GNN is expressed as:
>
> $$
> \text{Maximize:} \ \ \log{p(X_{obs})} = \int\log{ q(Z^L_{O},Z^L_{F}|X_{obs})\cdot{\frac{p{(Z^L_{O},Z^L_{F},X_{obs})}}{q{(Z^L_O,Z^L_F|X_{obs})}}}dZ} \geq \int q(Z^L_O, Z^L_F|X_{obs}) \log{\frac{p(X_{obs},Z^L_O,Z^L_F)}{q(Z^L_O,Z^L_F|X_{obs})}} dZ
> $$
> $\text{That is to maximize}$: $E_{q(Z^L_{O},Z^L_F|X_{obs})}\log{p(X_{obs}|Z^L_{O},Z^L_{F})} - D_{\text{KL}}[p(Z^L_O,Z^L_F|X_{obs}) \parallel p(Z^L_O,Z^L_F)]$
>
> where $E_{q(Z_{O},Z_F|X_{obs})}\log{p(X_{obs}|Z_{O}^L,Z_{F}^L)}$ represents the $\textbf{reconstruction loss}$.
>
> In principle, the SHT-GNN and VAE-based methods share conceptual similarities. However, the key distinctions between the SHT-GNN and VAE-based approaches in the handling of longitudinal data are as follows.
>
> $\textbf{1. Enhanced Representation Capability through Graph Neural Networks (GNNs)}$. In SHT-GNN, the encoding and decoding processes utilize graph neural networks (GNNs) instead of the multilayer perceptrons (MLPs) employed in standard VAEs. GNNs offer significantly stronger representation capabilities by leveraging the inherent data structure and facilitating temporal smoothing across non-independent observations, which is critical in longitudinal settings.
>
> $\textbf{2. Distinct Regularization Term for Non-Independent Observations}$. In VAE-based methods, the regularization term for the latent space distribution, $D_{\text{KL}}[q(z|x) \parallel p(z)]$, quantifies the independent distributions of individual observations. By contrast, SHT-GNN incorporates a KL divergence term, $D_{\text{KL}}[p(Z^L_O,Z^L_F|X_{obs}) \parallel p(Z^L_O,Z^L_F)]$, which measures the joint distribution of all observation representations. To account for dependencies within longitudinal data, SHT-GNN employs multi-layer temporal smoothing and introduces MADGap, enabling improved representation of the joint latent space distribution.
>
> In summary, the foundational intuition behind SHT-GNN lies in optimizing the reconstruction error during the data reconstruction process. Moreover, in the context of longitudinal data, SHT-GNN introduces temporal smoothing and MADGap to strike a balance between temporal consistency and the diversity of observation representations within the latent space.
>
> We have realized that the lack of straightforward presentation and theoretical intuitiveness in our paper has caused difficulties for readers. Therefore, in the revision of our paper, we will include all the aforementioned theoretical understandings as a remark within the article.

---

> ### Author Response · Authors · 2024-11-19
> **Weakness 2 and Question 1 (Part 1): Ignored baseline, Transformer-based baselines and real dataset supplementation**
>
> 1. $\textbf{Ignored baseline: “On the unreasonable effectiveness of feature propagation in learning on graphs with missing node features”}$
>
> Although this paper leverages Graph Neural Networks, it does not address longitudinal or dynamic time-series data. Instead, its primary focus is on static graph data, such as social interactions and traffic flows. While the study incorporates missing data imputation within a GNN framework, it is not designed for temporal data. Similar works include:
>
> - [1] Um D, Park J, Park S, et al. Confidence-based feature imputation for graphs with partially known features [J]. arXiv preprint arXiv:2305.16618, 2023.
>
> - [2] Telyatnikov L, Scardapane S. EGG-GAE: scalable graph neural networks for tabular data imputation [C]//International Conference on Artificial Intelligence and Statistics. PMLR, 2023: 2661-2676.
>
> These studies also focus on missing data imputation in static graph or tabular datasets. However, they do not address the challenges of imputing longitudinal or time-series data, which is the core problem our work seeks to tackle.

---

> > ### Author Response · Authors · 2024-11-19
> > **Question 2: Appendix Simulations and their reliance to existing studies**
> >
> > **1. Appendix Simulations and their reliance to existing studies**
> >
> > The simulations in the appendix follow well-established data generation formulations and temporal smoothing methodologies in longitudinal data research. Specifically, they draw on approaches described in:
> >
> > - **[1]** Binder H, Sauerbrei W, Royston P. Comparison between splines and fractional polynomials for multivariable model building with continuous covariates: a simulation study with continuous response [J]. *Statistics in Medicine*, 2013, 32(13): 2262-2277.
> >
> > - **[2]** Wu H, Zhang JT. *Nonparametric regression methods for longitudinal data analysis: mixed-effects modeling approaches* [M]. John Wiley \& Sons, 2006.
> >
> > Specifically, the generation of continuous response variables followed the approach outlined in [1]. The simulation formula proposed in [1] was designed to ensure that the generated data align with a predefined Structural Causal Model. This was achieved by employing a linear or nonlinear combination of covariates’ lower-order terms (e.g., square root or original values), higher-order terms (e.g., squared or cubed values), and interaction terms. In our work, the covariate dimensionality in the simulation was relatively high. To increase the complexity of the relationship between covariates and the response variable, the final simulation employed a combination of linear and nonlinear terms.
> >
> > Subsequently, the process of averaging observations within the same time window and adding noise followed the methodology described in [2].
> >
> > **2. Suggestion to rename the model to SHT-GNN**
> >
> > Thank you for your suggestion! We will adopt it and revise the model name to SH-GNN in our updated revision.

---

> > ### Author Response · Authors · 2024-11-20
> > **Weakness 2 and Question 1 (Part 2): Ignored baseline, Transformer-based baselines and real dataset supplementation**
> >
> > $\textbf{Transformer-based baselines and real dataset supplementation}$
> >
> > Recent cutting-edge methods employing transformers for longitudinal data imputation offer faster computation but share a significant limitation with RNN-based approaches. Specifically, these methods require alignment of observation steps across all subjects prior to computation. For instance, in the ADNI dataset, while most individuals have approximately 20–30 follow-ups, a considerable number have fewer than 10. Preparing data for these models necessitates padding or truncating observations to achieve uniformity in the number of observation steps. This preprocessing not only imposes additional computational overhead, but may also introduce bias into downstream model training. In contrast, our proposed method completely eliminates the need for such pre-processing. By facilitating information sharing across observation steps for different subjects, it effectively addresses the challenges associated with alignment and preprocessing, offering a more efficient and unbiased approach to longitudinal data imputation.
> >
> > In this revision, we have supplemented our analysis by incorporating results from transformer-based methods applied to various real-world datasets, alongside other state-of-the-art baselines. The supplementary baselines are:
> >
> > **[1]** Transformer: Ailing Zeng, Muxi Chen, Lei Zhang, and Qiang Xu. 2022. Are transformers effective for time series forecasting? arXiv preprint arXiv:2205.13504 (2022).
> >
> > **[2]**  GP-VAE: Fortuin V, Baranchuk D, Rätsch G, et al. Gp-vae: Deep probabilistic time series imputation[C]//International conference on artificial intelligence and statistics. PMLR, 2020: 1651-1661.
> >
> > **[3]**  SAITS: Du W, Côté D, Liu Y. Saits: Self-attention-based imputation for time series[J]. Expert Systems with Applications, 2023, 219: 119619.
> >
> > **[4]**  CTA: Wi H, Shin Y, Park N. Continuous-time Autoencoders for Regular and Irregular Time Series Imputation[C]//Proceedings of the 17th ACM International Conference on Web Search and Data Mining. 2024: 826-835.
> >
> > For each dataset, we introduce missing data under an MCAR mechanism with a missing ratio of 0.3. We subsequently validated the imputation performance using the Mean Absolute Error (MAE) metric.
> >
> > |             | AirQuality | Electricity | Energy    |
> > | ----------- | ---------- | ----------- | --------- |
> > | Transformer | 0.220      | 0.889       | 0.313     |
> > | GP-VAE      | 0.287      | 0.963       | 0.401     |
> > | CTA         | **0.196**  | **0.797**   | 0.205     |
> > | SAITS       | 0.207      | 0.894       | 0.301     |
> > | GRAPE       | 0.251      | 0.874       | 0.211     |
> > | IGRM        | 0.248      | 0.864       | 0.209     |
> > | **SHT-GNN** | 0.203      | 0.814       | **0.183** |
> >
> > For each dataset, we also applied MCAR with a missing ratio of 0.5, and subsequently validated the missing data imputation results using MAE.
> >
> > |             | AirQuality | Electricity | Energy    |
> > | ----------- | ---------- | ----------- | --------- |
> > | Transformer | 0.235      | 0.946       | 0.393     |
> > | GP-VAE      | 0.303      | 0.993       | 0.486     |
> > | CTA         | **0.203**  | **0.812**   | 0.227     |
> > | SAITS       | 0.220      | 0.943       | 0.351     |
> > | GRAPE       | 0.269      | 0.892       | 0.239     |
> > | IGRM        | 0.254      | 0.879       | 0.247     |
> > | **SHT-GNN** | 0.211      | 0.823       | **0.203** |
> >
> > where the AirQuality dataset consists of a single-object time series with 15 features and over 9,000 records. The Electricity dataset comprises long-term time series data for 370 subjects, containing a total of over 140,256 records, each characterized by a single-dimensional attribute. The Energy dataset captures energy usage from various electrical appliances, representing 110 subjects with more than 10,000 observations and a total of 27 features.
> >
> > On the AirQuality and Electricity datasets, the SHT-GNN method demonstrates inferior performance compared to RNN and VAE-based approaches. This indicates that the SHT-GNN model is not well-suited for long-term, single-object time series, which fall outside its intended application scenario.
> >
> > However, SHT-GNN achieves state-of-the-art performance on the Energy dataset. Unlike the AirQuality and Electricity datasets, the Energy dataset involves time series with multiple subjects and multidimensional features. SHT-GNN leverages its ability to effectively borrow information across features, which significantly enhances its performance, particularly in the context of irregular longitudinal data imputation.

---

> > > ### Comment · Reviewer_iXB5 · 2024-11-26
> > >
> > > Thank you for adding the experiments. Regarding the comment "SHT-GNN model is not well-suited for long-term, single-object time series", in practice, how do you determine if a time series qualifies as "long-term, single-object"? This determination is crucial for deciding whether to use SHT-GNN or transformer-based models like CTA.

---

> > ### Comment · Reviewer_iXB5 · 2024-11-26
> >
> > Although these methods focus on missing data imputation in static graphs, they could still serve as interesting baselines for imputing longitudinal or time-series data. This approach would involve ignoring the temporal relations and imputing data for each timestamp independently.

---

> ### Comment · Reviewer_iXB5 · 2024-11-26
>
> Thanks the authors for the detailed response. I have a few additional questions, but overall I will maintain my current rating and recommendation.

---

### Meta-Review · Area_Chair_FMgi · 2024-12-16

**Metareview:**

This work proposed a GNN model for longitudinal data imputation, where the observations, covariates, temporal and subject information are jointly considered in a heterogeneous graph. The reviewers agree that the paper addressed an important problem in dealing with longitudinal data, and that the revision has improved the empirical evaluation. The reviewers share the sentiment that the overall novelty can be better clarified to position this work in the literature. Despite the authors made efforts in explaining the mechanism based on graph VAE, the current writing lacks motivations and justifications from a formal perspective, and on how the authors' approach addresses limitations of existing methods. The generalizability to other types of missing data and more general data types should be thoroughly explained.

**Additional Comments On Reviewer Discussion:**

All five reviewers participated in the post-rebuttal discussion.

A shared criticism is that the submission should be better motivated, e.g. from theoretical perspective, and with its novelty clarified among the existing GNN-based imputation methods. The authors provided some explanation from graph VAE, which is relevant but not tightly aligned with the core mechanisms applied. Moreover, this should be addressed in the early sections to motivate the construction of the GNN.

The authors mentioned that three reviewers are not responsive. This is not true. The mentioned reviewers have interacted with the author
s in the rebuttal phase.

The authors' rebuttal is unfortunately not well organized to address the key issues raised by the reviewers.  The same contents and requests are posted multiple times. One reviewer (iXB5) has raised additional questions which are somehow ignored by the authors.

---

### Decision · Program_Chairs · 2025-01-22

Reject